# A principle of economy predicts the functional architecture of grid cells

Xue-Xin Wei[1], Jason Prentice[2], Vijay Balasubramanian[3,4]*

[1]Department of Psychology, University of Pennsylvania, Philadelphia, United States; [2]Princeton Neuroscience Institute, Princeton University, Princeton, United States; [3]Department of Physics, University of Pennsylvania, Philadelphia, United States; [4]Department of Neuroscience, University of Pennsylvania, Philadelphia, United States

**Abstract** Grid cells in the brain respond when an animal occupies a periodic lattice of 'grid fields' during navigation. Grids are organized in modules with different periodicity. We propose that the grid system implements a hierarchical code for space that economizes the number of neurons required to encode location with a given resolution across a range equal to the largest period. This theory predicts that (i) grid fields should lie on a triangular lattice, (ii) grid scales should follow a geometric progression, (iii) the ratio between adjacent grid scales should be $\sqrt{e}$ for idealized neurons, and lie between 1.4 and 1.7 for realistic neurons, (iv) the scale ratio should vary modestly within and between animals. These results explain the measured grid structure in rodents. We also predict optimal organization in one and three dimensions, the number of modules, and, with added assumptions, the ratio between grid periods and field widths.

## Introduction

How does the brain represent space? *Tolman (1948)* suggested that the brain must have an explicit neural representation of physical space, a *cognitive map*, that supports higher brain functions such as navigation and path planning. The discovery of place cells in the rat hippocampus (*O'Keefe, 1976*; *O'Keefe and Nadel, 1978*) suggested one potential locus for this map. Place cells have spatially localized firing fields which reorganize dramatically when the environment changes (*Leutgeb et al., 2005*). Another potential locus for the cognitive map of space has been uncovered in the main input to hippocampus, a structure known as the medial entorhinal cortex (MEC) (*Figure 1*, *Fyhn et al., 2004*; *Hafting et al., 2005*). When rats freely explore a two-dimensional open environment, individual 'grid cells' in the MEC display spatial firing fields that form a periodic triangular grid which tiles space (*Figure 1A*). It is believed that grid fields provide relatively rigid coordinates on space based partly on self-motion and partly on environmental cues (*Moser et al., 2008*). The scale of grid fields varies systematically along the dorso–ventral axis of the MEC (*Figure 1A*) (*Hafting et al., 2005*; *Barry et al., 2007*; *Stensola et al., 2012*). Recently, it was shown that grid cells are organized in discrete modules within which cells share the same orientation and periodicity but vary randomly in phase (*Barry et al., 2007*; *Stensola et al., 2012*).

How does the grid system represent spatial location and what function does the modular variation in grid scale serve? Here, we propose that the grid system provides a hierarchical representation of space where fine grids provide precise location and coarse grids resolve ambiguity, and that the grids are organized to minimize the number of neurons required to achieve the behaviorally necessary spatial resolution across a spatial range equal in size to the period of the largest grid module. Our analyses thus assume that there is a behaviorally defined maximum range over which a fixed grid represents locations. Our hypotheses, together with general assumptions about tuning curve shape and decoding mechanism, explain the triangular lattice structure of two-dimensional grid cell firing maps and predict a geometric

*For correspondence:
vijay@physics.upenn.edu

**Competing interests:** The authors declare that no competing interests exist.

**eLife digest** In the 1930s, neuroscientists studying how rodents find their way through a maze proposed that the animals could construct an internal map of the maze inside their heads. The map was thought to enable the animals to navigate between familiar locations and also to identify shortcuts and alternative routes whenever familiar ones were blocked.

In the 1960s, recordings of electrical activity in the rat brain provided the first clues as to which nerve cells form this spatial map. In a region of the brain called the hippocampus, nerve cells called 'place cells' are active whenever the rat finds itself in a specific location. However, place cells alone are not able to support all types of navigation. Some spatial tasks also require cells in a region of the brain called the medial entorhinal cortex (MEC), which supplies most of the information that the hippocampus receives.

Cells in the MEC called 'grid cells' represent two-dimensional space as a repeating grid of triangles. A given grid cell is activated if the animal is located at a particular distance and angle away from the center of any of these triangles. The size of the triangles in these grids varies systematically throughout the MEC. Individual grid cells at one end of the structure encode space in finer detail than grid cells at the opposite end.

Wei et al. have now used mathematical modeling to explore how grid cells are organized. The model assumes that the brain seeks to encode space at whatever resolution an animal requires using as few nerve cells as possible. The model successfully reproduces several known features of grid cells, including the triangular shape of the grid, and the fact that the size of the triangles increases in steps of a specific size across the MEC.

In addition to providing a mathematical basis for the way that grid cells are organized in the brain, the model makes a number of testable predictions. These include predictions of the number of grid cells in the rat brain, as well as the pattern that grid cells adopt in three-dimensions: a question that is currently being studied in bats. Wei et al.'s findings suggest that the code used by the grid to represent space is an analog of a decimal number system—except that space is not subdivided by factors of 10 to form decimal 'digits', but by a quantity related to a famous constant in the field of mathematics called Euler's number.

progression of grid scales. Crucially, the theory further predicts that the ratio of adjacent grid scales will be modestly variable within and between animals with a mean in the range 1.4–1.7 depending on the assumed decoding mechanism used by the brain. With additional assumptions the theory also predicts that the ratio between grid scale and individual grid field widths should lie in the same range. These predictions naturally explain the structural parameters of grid cell modules measured in rodents (*Barry et al., 2007*; *Giocomo et al., 2011a*; *Stensola et al., 2012*). Our results follow from general principles, and thus, we expect similar organization of the grid system in other species. The theory makes further predictions including: (a) the number of grid scales necessary to support navigation over typical behavioral distances (i.e., a logarithmic relation between number of modules and navigational range), (b) possible deficits in spatial behavior that will obtain upon inactivating specific grid modules, (c) the structure of one- and three-dimensional grids that will be relevant to navigation in, for example, bats (*Yartsev et al., 2011*), (d) an estimate of the number of grid cells we expect in the mEC. Remarkably, in a simple decoding scheme, the scale ratio in an *n*-dimensional environment is predicted to be close to $\sqrt[n]{e}$.

As we will explain, our results and their apparent experimental confirmation in *Stensola et al. (2012)*, suggest that the grid system implements a two-dimensional neural analog of a base-b number system. This provides an intuitive and powerful metaphor for interpreting the representation of space in the entorhinal cortex.

## Results

### The set-up

The key features of the grid system in the MEC are schematized in *Figure 1A*. Grid cells are organized in modules, and cells within a module share a common lattice organization of their firing fields

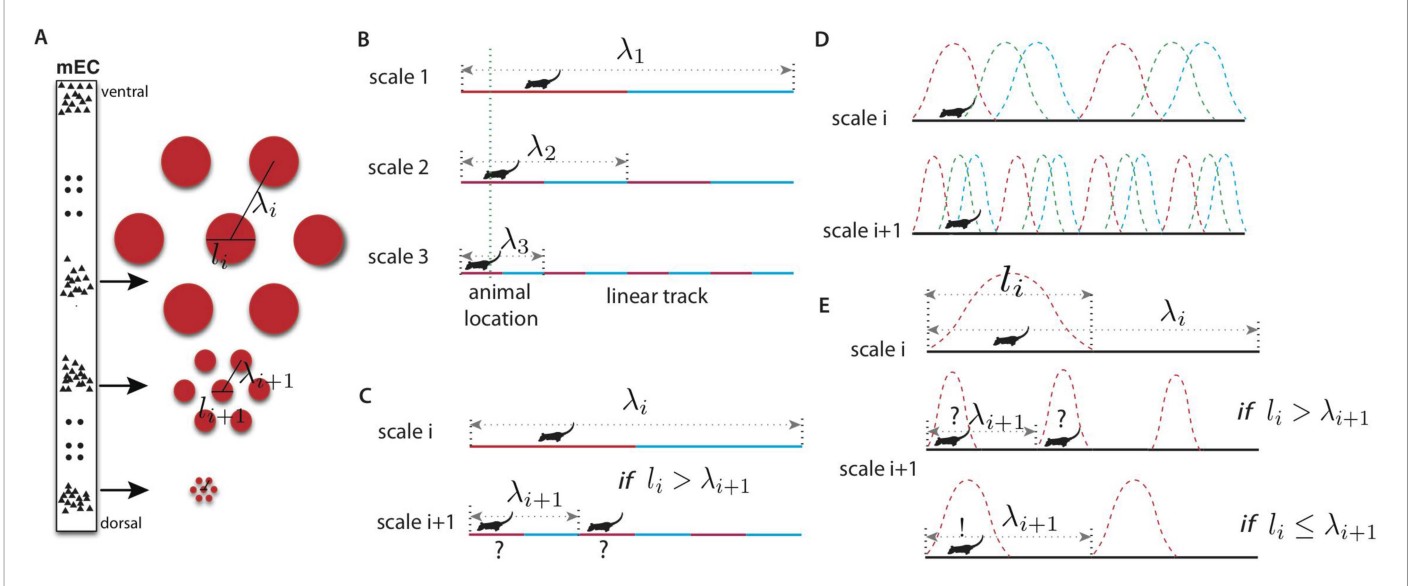

**Figure 1**. Representing place in the grid system. (**A**) Grid cells (small triangles) in the medial entorhinal cortex (MEC) respond when the animal is in a triangular lattice of physical locations (red circles) (*Fyhn et al., 2004*; *Hafting et al., 2005*). The scale of periodicity (the 'grid scale', $\lambda_i$) and the size of the regions evoking a response above a noise threshold (the 'grid field width', $l_i$) vary modularly along the dorso-ventral axis of the MEC (*Hafting et al., 2005*). Grid cells within a module vary in the phase of their spatial response, but share the same period and grid orientation (in two dimensions) (*Stensola et al., 2012*). (**B**) A simplified binary grid scheme for encoding location along a linear track. At each scale ($\lambda_i$) there are two grid cells (red vs blue firing fields). The periodicity and grid field widths are halved at each successive scale. (**C**) The binary scheme in (**B**) is ambiguous if the grid field width at scale $i$ exceeds the grid periodicity at scale $i + 1$. For example, if the grid fields marked in red respond at scales $i$ and $i + 1$, the animal might be in either of the two marked locations. (**D**) The grid system is composed of discrete modules, each of which contains neurons with periodic tuning curves, and varying phase, in space. (**E**) For a simple winner-take-all decoder of the grids in panel **D**, decoded position will be ambiguous unless $l_i \leq \lambda_{i + 1}$, analogously to panel **C** (see text). Variants of this limitation occur in other decoding schemes.

(*Barry et al., 2007*; *Stensola et al., 2012*). These lattices have periods $\lambda_1 > \lambda_2 > \cdots \lambda_m$, measured as the distance between nearest neighbor firing fields. It will prove convenient to define 'scale factors' $r_i = \lambda_i / \lambda_{i+1}$ relating the periods of adjacent scales. In each module, the grid firing fields (i.e., the connected spatial regions that evoke firing) are compact (with a diameter denoted $l_i$) after thresholding for activity above the noise level (see, e.g., *Hafting et al., 2005*). Within any module, grid cells have a variety of spatial phases so that at least one cell will respond at any physical location (*Figure 1B,D*). Grid modules with smaller field widths $l_i$ provide more local spatial information than those with larger scales. However, this increased spatial precision comes at a cost: the correspondingly smaller periodicity $\lambda_i$ of these modules leads to increased ambiguity since there are more grid periods within a given spatial region (e.g., see scale 3 in the schematic one-dimensional grid in *Figure 1B,D*). By contrast, modules with large periods and field widths have less spatial precision, but also less ambiguity (e.g., in scale 1 in *Figure 1B* the red cell has only one firing field in the environment and hence no ambiguity).

We propose that the entorhinal cortex exploits this trade-off to implement a hierarchical representation of space where large scales resolve ambiguity and small scales provide precision. Consistently with existing data for one- and two-dimensional grids (*Barry et al., 2007*; *Brun et al., 2008*; *Stensola et al., 2012*), we will take the largest grid period $\lambda_1$ to be comparable to the range over which space is represented unambiguously by a fixed grid without remapping (*Fyhn et al., 2007*). (An alternative view, that the range might greatly exceed the largest period, is addressed in the 'Discussion'.) The spatial resolution of such a grid can be measured by comparing the range of spatial representation set by the largest period $\lambda_1$ to the precision (related to the smallest grid field width $l_m$) to quantify how many distinct spatial 'bins' can be resolved. We will assume that the required resolution is set by the animal's behavioral requirements.

## Intuitions from a simplified model

What are the advantages of a multi-scale, hierarchical representation of physical location? Consider an animal living in an 8 m linear track and requiring spatial precision of 1 m to support its behavior. To develop intuition, consider a simple model where location is represented in the animal's brain by reliable neurons with rectangular firing fields (e.g., *Figure 1B*). The animal could achieve the required resolution in a *place coding* scheme by having eight neurons tuned to respond when the animal is in 1 m wide, non-overlapping regions (see [*Fiete et al., 2008*] for a related comparison between grid and place cells). Consider an alternative, the idealized *grid coding* scheme in *Figure 1B*. Here, the two neurons at the largest scale ($\lambda_1$) have 4 m wide tuning curves so that their responses just indicate the left and right halves of the track. The pairs of neurons at the next two scales have grid field widths of 2 m and 1 m respectively, and proportionally shorter periodicities as well. These pairs successively localize the animal into 2 m and 1 m bins. All told only six neurons are required, less than in the place coding scheme. This suggests that grid schemes that integrate multiple scales of representation can encode space more efficiently, that is, with fewer neural resources. In the sensory periphery, there is evidence of selection for more efficient circuit architectures (e.g., *Simoncelli and Olshausen, 2001*). If similar selection operates in cortex, the experimentally measured grid architecture should be predicted by maximizing the efficiency of the grid system given a behaviorally determined range and resolution. Thus, we seek to predict the key structural parameters of the grid system—the ratios $r_i = \lambda_i/\lambda_{i+1}$ relating adjacent scales (which need not be equal).

The need to avoid spatial ambiguity constrains the ratios $r_i$. Again in our simple model, consider *Figure 1C* where the cells with the grid fields marked in red respond at scales $i$ and $i + 1$. Then the animal might be in either of the two marked locations. Avoiding ambiguity requires that $\lambda_{i+1}$, the period at scale $i + 1$, must exceed $l_i$, the grid field width at scale $i$. Variants of this condition will recur in the more realistic models that we will consider. Theoretically, one could resolve the ambiguity in *Figure 1C* by combining the responses of more grid modules, provided they have mutually incommensurate periods (*Fiete et al., 2008*; *Sreenivasan and Fiete, 2011*). However, anatomical evidence suggests that contiguous subsets of the mEC along the dorso–ventral axis project topographically to the hippocampus (*Van Strien et al., 2009*). While there is evidence that hippocampal place cells are not formed and maintained by grid cell inputs alone (*Bush et al., 2014*; *Sasaki et al., 2015*), for each of these restricted projections to represent a well-defined spatial map, ambiguities like the one in *Figure 1C* should be resolved at each scale. The hierarchical position encoding schemes that we consider below embody this observation by seeking to reduce position ambiguity at each scale, given the responses at larger scales.

## Efficient grid coding in one dimension

How should the grid system be organized to minimize the resources required to represent location unambiguously with a given resolution? Consider a one-dimensional grid system that develops when an animal runs on a linear track. As described above, the $i$th module is characterized by a period $\lambda_i$, while the ratio of adjacent periods is $r_i = \lambda_i/\lambda_{i+1}$. Within any module, grid cells have periodic, bumpy response fields with a variety of spatial phases so that at least one cell responds at any physical location (*Figure 1D*). If $d$ cells respond above the noise threshold at each point, the number of grid cells $n_i$ in module $i$ will be $n_i = d\lambda_i/l_i$. We will take $d$, the *coverage factor*, to be the same in each module. In terms of these parameters, the total number of grid cells is $N = \sum_{i=1}^{m} n_i = \sum_{i=1}^{m} d\frac{\lambda_i}{l_i}$, where $m$ is the number of grid modules. How should such a grid be organized to minimize the number of grid cells required to achieve a given spatial resolution? The answer might depend on how the brain decodes the grid system. Hence, we will consider decoding methods at extremes of decoding complexity and show that they give similar answers for the optimal grid.

### Winner-take-all decoder

First imagine a decoder which considers the animal as localized within the grid fields of the most responsive cell in each module (*Coultrip et al., 1992*; *Maass, 2000*). A simple 'winner-take-all' (WTA) scheme of this kind can be easily implemented by neural circuits where lateral inhibition causes the influence of the most responsive cell to dominate. A maximally conservative decoder ignoring all information from other cells and from the shape of the tuning curve (illustrated in *Figure 1E*) could then take uncertainty in spatial location to be equal to $l_i$. The smallest interval that can be resolved in this way will be $l_m$. We therefore quantify the resolution of the grid system (the number of spatial bins

that can be resolved) as the ratio of the largest to the smallest scale, $R_1 = \lambda_1 / l_m$, which we assume to be large and fixed by the animal's behavior. In terms of scale factors $r_i = \lambda_i / \lambda_{i+1}$, we can write the resolution as $R_1 = \prod_{i=1}^{m} r_i$, where we also defined $r_m = \lambda_m / l_m$. As in our simplified model above, unambiguous decoding requires that $l_i \leq \lambda_{i+1}$ (**Figure 1C,E**), or, equivalently, $\frac{\lambda_i}{l_i} \geq r_i$. To minimize $N = d \sum_i \lambda_i / l_i$, all the $\frac{\lambda_i}{l_i}$ should be as small as possible; so this fixes $\frac{\lambda_i}{l_i} = r_i$. Thus, we are reduced to minimizing the sum $N = d \sum_{i=1}^{m} r_i$ over the parameters $r_i$, while fixing the product $R_1 = \prod_i r_i$. Because this problem is symmetric under permutation of the indices $i$, the optimal $r_i$ turn out to all be equal, allowing us to set $r_i = r$ (Optimizing the grid system: winner-take-all decoder, 'Materials and methods'). This is our first prediction: (1) the ratios between adjacent periods will be constant. The constraint on resolution then gives $m = \log_r R_1$, so that we seek to minimize $N(r) = d\, r \log_r R_1$ with respect to $r$: the solution is $r = e$ (Optimizing the grid system: winner-take-all decoder, 'Materials and methods', and panel B of Figure 5 in Optimizing the grid system: probabilistic decoder, 'Materials and methods'). This gives a second prediction: (2) the ratio of adjacent grid periods should be close to $r = e$. Therefore, for each scale $i$, $\lambda_i = e\,\lambda_{i+1}$ and $\lambda_i = e l_i$. This gives a third prediction: (3) the ratio of the grid period and the grid field width will be constant across modules and be close to the scale ratio.

More generally, in winner-take-all decoding schemes, the local uncertainty in the animal's location in grid module $i$ will be proportional to the grid field width $l_i$. The proportionality constant will be a function $f(d)$ of the coverage factor $d$ that depends on the tuning curve shape and neural variability. Thus, the uncertainty will be $f(d)l_i$. Unambiguous decoding at each scale requires that $\lambda_{i+1} \geq f(d)l_i$. The smallest interval that can be resolved in this way will be $f(d)l_m$, and this sets the positional accuracy of the decoding scheme. Finally, we require that $\lambda_1 > L$, where $L$ is a scale big enough to ensure that the grid code resolves positions over a sufficiently large range. Behavioral requirements fix the required positional accuracy and range. The optimal grid satisfying these constraints is derived in Optimizing the grid system: winner-take-all decoder, 'Materials and methods'. Again, the adjacent modules are organized in a geometric progression and the ratio between adjacent periods is predicted to be $e$. However, the ratio between the grid period and grid field width in each module depends on the specific model through the function $f(d)$. Thus, within winner-take-all decoding schemes, the constancy of the scale ratio, the value of the scale ratio, and the constancy of the ratio of grid period to field width are parameter-free predictions, and therefore furnish tests of theory. If the tests succeed, $f(d)$ can be matched to data to constrain possible mechanisms used by the brain to decode the grid system.

## Probabilistic decoder

What do we predict for a more general, and more complex, decoding scheme that optimally pools all the information available in the responses of noisy neurons within and between modules? Statistically, the best we can do is to use all these responses, which may individually be noisy, to find a probability distribution over physical locations that can then inform subsequent behavioral decisions (**Figure 2**). Thus, the population response at each scale $i$ gives rise to a likelihood function over location $P(x|i)$, which will have the same periodicity $\lambda_i$ as the individual grid cells' firing rates (**Figure 2A**). This likelihood explicitly captures the uncertainty in location given the tuning and noise characteristics of the neural population in the module $i$. Because there are at least scores of neurons in each grid module (**Stensola et al., 2012**) $P(x|i)$ can be approximated as a periodic sum of Gaussians without making restrictive assumptions about the shapes of the tuning curves of individual grid cells, or about the precision of their periodicity, so long as the variability of individual neurons is weakly correlated and homogeneous. For example, even though individual grid cells can have somewhat different firing rates in each of their firing fields, this spatial heterogeneity will be smoothed in the posterior over the full population of cells, leading to much more accurate periodicity. In other words, individual grid cells show both spiking noise and 'noise' due to heterogeneity and imperfect periodicity of the firing rate maps. Both these forms of variability are smoothed by averaging over the population, provided, as we will assume, that there are enough cells and noise is not too correlated between cells.

The standard deviations of the peaks in $P(x|i)$, which we call $\sigma_i$, depend on the tuning curve shape and response noise of individual grid cells, and will decrease as the coverage factor $d$ increases. To have even coverage of space, the number of grid phases, and thus grid cells in a module, must be uniformly distributed so that equally reliable posterior distributions can be formed at each point in the unit cell of the module response. This requires that the number of cells (and phases) in the module should be proportional to the ratio $\frac{\lambda_i}{\sigma_i}$. Summing over modules, the total number of grid cells will be

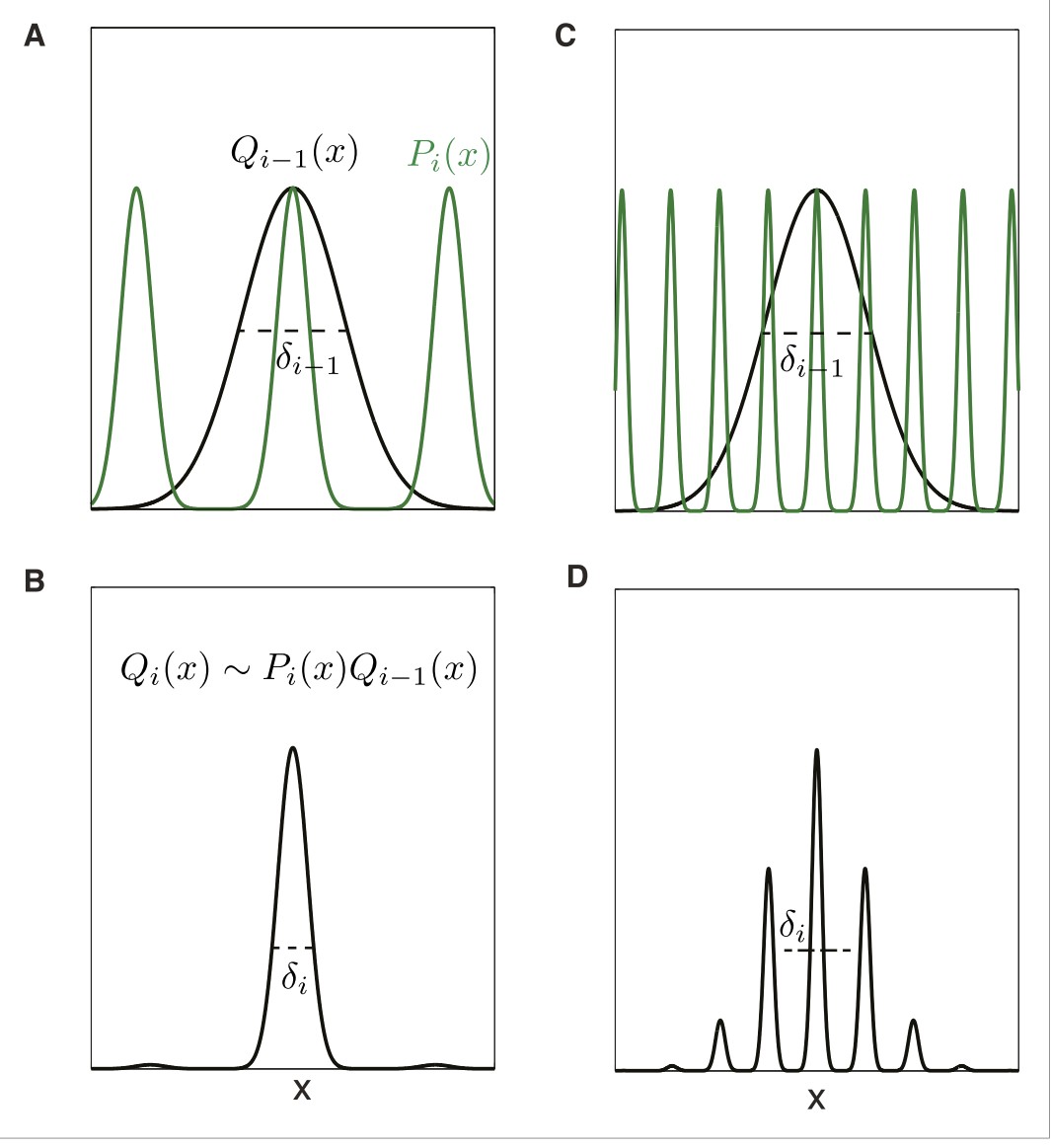

**Figure 2.** Trade-off between precision and ambiguity in the probabilistic decoder. (**A**) The probability of position $x$ given the responses of all grid cells at scales larger than module $i$ is described by the distribution $Q_{i-1}(x)$ (black curve), and the uncertainty in position is given by the standard deviation $\delta_{i-1}$. The probability of position given just the responses in module $i$ will be a periodic function $P_i(x)$ (green curve). (**B**) The probability distribution over position $x$ after combining module $i$ with all larger scales is $Q_i(x) \sim P_i(x)Q_{i-1}(x)$ and has reduced uncertainty $\delta_i$. (**C**) Precision can be improved by increasing the scale factor, thereby narrowing the peaks of $P_i(x)$. However, the periodicity shrinks as well, increasing ambiguity. (**D**) The distribution over position $Q_i(x)$ from combining the modules shown in **C**. Ambiguity from the secondary peaks leads to an overall uncertainty $\delta_i$ larger than in **B**, despite the improved precision from the narrower central peak.

$N \propto \sum_{i=1}^{m} \frac{\lambda_i}{\sigma_i}$. The composite posterior given all $m$ scales and a uniform prior over positions, $Q_m(x)$, will be given by the product $Q_m(x) \propto \Pi_{i=1}^{m} P(x|i)$, assuming independent response noise across scales (*Figure 2B*). The animal's overall uncertainty about its position depends on the standard deviation $\delta_m$ of the composite posterior distribution $Q_m(x)$. Setting $\delta_0$ to be the uncertainty in location without using any grid responses at all, we can quantify resolution as $R = \delta_0/\delta_m$.

In this framework, there is a precision-ambiguity trade-off controlled by the scale factors $r_i$. The larger these ratios, the more rapidly grid field widths shrink in successive modules, thus increasing precision and reducing the number of modules, and hence grid cells, required to achieve a given

resolution. However, if the periods of adjacent scales shrink too quickly, the composite posterior $Q_i(x)$ will develop prominent side-lobes (*Figure 2C,D*) making decoding ambiguous, as reflected in a large standard deviation $\delta_i$ of the composite posterior distribution (*Figure 2B,D*). This ambiguity could be avoided by shrinking the width of $Q_{i-1}(x)$—however, this would require increasing the number of neurons $n_1, \cdots n_{i-1}$ in the modules $1, \cdots i - 1$. Ambiguity can also be avoided by having a smaller scale ratio (so that the side lobes of the posterior $P(x|i)$ of module $i$ do not penetrate the central lobe of the composite posterior $Q_{i-1}(x)$ of modules $1, \cdots i–1$. But reducing the scale ratios to reduce ambiguity increases the number of modules necessary to achieve the required resolution, and hence increases the number of grid cells. This sets up a trade-off—increasing the scale ratios reduces the number of modules to achieve a fixed resolution but requires more neurons in each module; reducing the scale ratios permits the use of fewer grid cells in each module, but increases the number of required modules. Optimizing this trade-off (analytical and numerical details in 'Materials and methods' and Figure 5) predicts: (1) a constant scale ratio between the periods of each grid module, and (2) an optimal ratio $\approx 2.3$, slightly smaller than, but close to the winner-take-all value, $e$.

Why is the predicted scale factor based on the probabilistic decoder somewhat smaller than the prediction based on the winner-take-all analysis? In the probabilistic analysis, when the likelihood is combined across modules, there will be side lobes arising from the periodic peaks of the likelihood derived from module $i$ multiplying the tails of the Gaussian arising from the previous modules. These side lobes increase location ambiguity (measured by the standard deviation $\delta_i$ of the overall likelihood). Reducing the scale factor reduces the height of side lobes because the secondary peaks from module $i$ move further into the tails of the Gaussian derived from the previous modules. Thus, conceptually, the optimal probabilistic scale factor is smaller than the winner-take-all case in order to suppress side lobes that arise in the combined likelihood across modules (*Figure 2*). Such side lobes were absent in the winner-take-all analysis, which thus permits a more aggressive (larger) scale ratio that improves precision, without being penalized by increased ambiguity. The theory also predicts a fixed ratio between grid period $\lambda_i$ and posterior likelihood width $\sigma_i$. However, the relationship between $\sigma_i$ and the more readily measurable grid field width $l_i$ depends on a variety of parameters including the tuning curve shape, noise level, and neuron density.

## General grid coding in two dimensions

How do these results extend to two dimensions? Let $\lambda_i$ be the distance between nearest neighbor peaks of grid fields of width $l_i$ (*Figure 3*). Assume in addition that a given cell responds on a lattice whose vertices are located at the points $\lambda_i (n\mathbf{u} + m\mathbf{v})$, where $n$, $m$ are integers and $\mathbf{u}$, $\mathbf{v}$ are linearly independent vectors generating the lattice (*Figure 3A*). We may take $\mathbf{u}$ to have unit length ($|\mathbf{u}| = 1$) without loss of generality, however $|\mathbf{v}| \neq 1$ in general. It will prove convenient to denote the components of $\mathbf{v}$ parallel and perpendicular to $\mathbf{u}$ by $v_{||}$ and $v_{\perp}$, respectively (*Figure 3A*). The two numbers $v_{||}$, $v_{\perp}$ quantify the geometry of the grid and are additional parameters that we may optimize over: this is a primary difference from the one-dimensional case. We will assume that $v_{||}$ and $v_{\perp}$ are independent of scale; this still allows for relative rotation between grids at different scales. At each scale, grid cells have different phases so that at least one cell responds at each physical location. The minimal number of phases required to cover space is computed by dividing the area of the unit cell of the grid ($\lambda_i^2 ||\mathbf{u} \times \mathbf{v}|| = \lambda_i^2 |v_{\perp}|$) by the area of the grid field. As in the one-dimensional case, we define a coverage factor $d$ as the number of neurons covering each point in space, giving for the total number of neurons $N = d |v_{\perp}| \sum_i (\lambda_i / l_i)^2$.

As before, consider a situation where grid fields thresholded for noise lie completely within compact regions and assume a simple decoder which selects the most activated cell and does not take tuning curve shape into account (*Coultrip et al., 1992*; *Maass, 2000*; *de Almeida et al., 2009*). In such a model, each scale $i$ simply serves to localize the animal within a circle of diameter $l_i$. The spatial resolution is summarized by the square of the ratio of the largest scale $\lambda_1$ to the smallest scale $l_m$: $R_2 = (\lambda_1 / l_m)^2$. In terms of the scale factors $\tilde{r}_i = \lambda_i / \lambda_{i+1}$, we write $R_2 = \prod_{i=1}^{m} \tilde{r}_i^2$, where we also define $\tilde{r}_m = \lambda_m / l_m$. To decode the position of an animal unambiguously, each cell at scale $i + 1$ should have at most one grid field within a region of diameter $l_i$. We therefore require that the shortest lattice vector of the grid at scale $i$ has a length greater than $l_{i-1}$, in order to avoid ambiguity (*Figure 3B*). We wish to minimize $N$, which will be convenient to express as $N = d |v_{\perp}| \sum_i \tilde{r}_i^2 (\lambda_{i+1} / l_i)^2$. There are two kinds of contributions here to the number of neurons—the factors $\tilde{r}_i^2$ are constrained by the overall resolution of the grid,

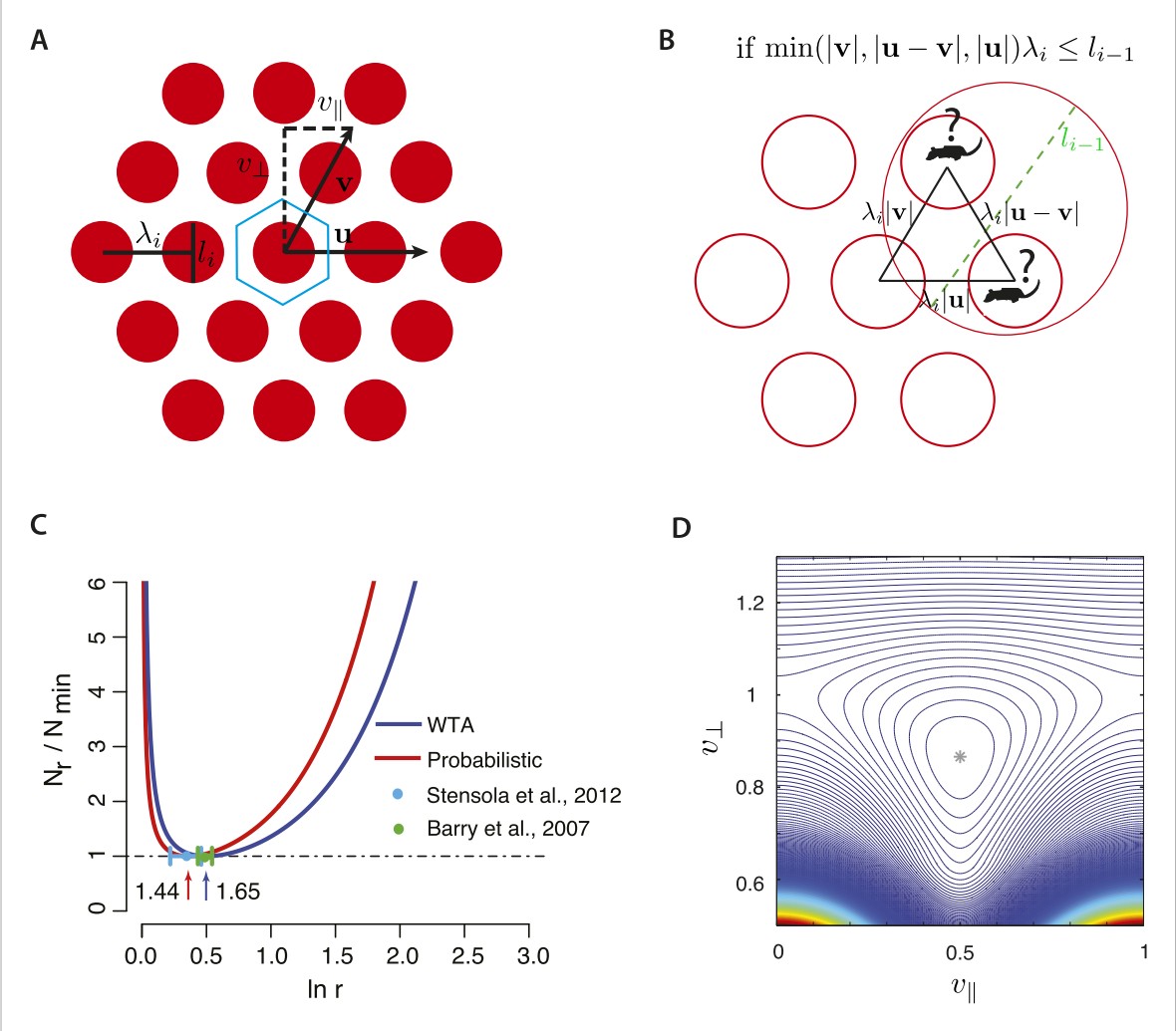

**Figure 3**. Optimizing two-dimensional grids. (**A**) A general two-dimensional lattice is parameterized by two vectors **u** and **v** and a periodicity parameter $\lambda_i$. Take **u** to be a unit vector, so that the spacing between peaks along the **u** direction is $\lambda_i$, and denote the two components of **v** by $v_{\parallel}$, $v_{\perp}$. The blue-bordered region is a fundamental domain of the lattice, the largest spatial region that can be unambiguously represented. (**B**) The two-dimensional analog of the ambiguity in *Figure 1C,E* for the winner-take-all decoder. If the grid fields in scale $i$ are too close to each other relative to the size of the grid field of scale $i-1$ (i.e., $l_{i-1}$), the animal might be in one of several locations. (**C**) The optimal ratio $r$ between adjacent scales in a hierarchical grid system in two dimensions for a winner-take-all decoding model (blue curve, WTA) and a probabilistic decoder (red curve). $N_r$ is the number of neurons required to represent space with resolution $R$ given a scaling ratio $r$, and $N_{min}$ is the number of neurons required at the optimum. In both decoding models, the ratio $N_r/N_{min}$ is independent of resolution, $R$. For the winner-take-all model, $N_r$ is derived analytically, while the curve for the probabilistic model is derived numerically (details in Optimizing the grid system: winner-take-all decoder and Optimizing the grid system: probabilistic decoder, 'Materials and methods'). The winner-take-all model predicts $r = \sqrt{e} \approx 1.65$, while the probabilistic decoder predicts $r \approx 1.44$. The minima of the two curves lie within each others' shallow basins, predicting that some variability of adjacent scale ratios is tolerable within and between animals. The green and blue bars represent a standard deviation of the scale ratios of the period ratios between modules measured in *Barry et al. (2007)*; *Stensola et al. (2012)*. (**D**) Contour plot of normalized neuron number $N/N_{min}$ in the probabilistic decoder, as a function of the grid geometry parameters $v_{\perp}$, $v_{\parallel}$ after minimizing over the scale factors for fixed resolution $R$. As in *Figure 3C*, the normalized neuron number is independent of $R$. The spacing between contours is 0.01, and the asterisk labels the minimum at $v_{\parallel} = 1/2$, $v_{\perp} = \sqrt{3}/2$; this corresponds to the triangular lattice.

while, as we will see, the combination $|v_{\perp}|(\lambda_{i+1}/l_i)^2$ measures a packing density of discs placed on the grid lattice. This suggests that we should separate the minimization of neuron number into first optimizing the lattice and then optimizing ratios. After doing so, we can check that the result is the global optimum.

To obtain the optimal lattice geometry, we can ignore the resolution constraint, as it depends only on the scale factors and not the grid geometry. We may then exploit an equivalence between our

optimization problem and the optimal circle-packing problem. To see this connection, consider placing disks of diameter $l_i$ on each vertex of the grid at scale $i + 1$. In order to avoid ambiguity, all points of the grid $i + 1$ must be separated by at least $l_i$: equivalently, the disks must not overlap. The density of disks is proportional to $l_i^2/(\lambda_{i+1}^2|v_\perp|)$, which is proportional to the reciprocal of each term in $N$. Therefore, *minimizing* neuron number amounts to *maximizing* the packing density; and the no-ambiguity constraint requires that the disks do not overlap. This is the optimal circle packing problem, and its solution in two dimensions is known to be the triangular lattice (*Thue, 1892*), so $v_{||} = 1/2$ and $v_\perp = \sqrt{3}/2$. Furthermore, the grid spacing should be as small as allowed by the no-ambiguity constraint, giving $\lambda_{i+1} = l_i$.

We have now reduced the problem to minimizing $N = \frac{d\sqrt{3}}{2}\sum_i \tilde{r}_i^2$, over the scale factors $\tilde{r}_i$, while fixing the resolution $R_2$. This optimization problem is mathematically the same as in one dimension if we formally set $r_i \equiv \tilde{r}_i^2$. This gives the optimal ratio $\tilde{r}_i^2 = e$ for all $i$ (*Figure 3C*). We conclude that in two dimensions, the optimal ratio of neighboring grid periodicities is $\sqrt{e} \approx 1.65$ for the simple winner-take-all decoding model, and the optimal lattice is triangular.

The optimal probabilistic decoding model from above can also be extended to two dimensions with the posterior distributions $P(x|i)$ becoming sums of Gaussians with peaks on the two-dimensional lattice. In analogy with the one-dimensional case, we then derive a formula for the resolution $R_2 = \lambda_1/\delta_m$ in terms of the standard deviation $\delta_m$ of the posterior given all scales. The quantity $\delta_m$ may be explicitly calculated as a function of the scale factors $\tilde{r}_i$ and the geometric factors $v_{||}$, $v_\perp$, and the minimization of neuron number may then be carried out numerically (Optimizing the grid system: probabilistic decoder, 'Materials and methods'). In this approach, the optimal scale factor turns out to be $\tilde{r}_i \approx 1.44$ (*Figure 3C*), and the optimal lattice is again triangular (*Figure 3D*). Attractor network models of grid formation readily produce triangular lattices (*Burak and Fiete, 2009*); our analysis suggests that this architecture is functionally beneficial in reducing the required number of neurons.

Even though our two decoding strategies lie at extremes of complexity (one relying just on the most active cell at each scale and another optimally pooling information in the grid population) their respective 'optimal intervals' substantially overlap (*Figure 3C*; see Figure 5B in 'Materials and methods' for the one-dimensional case). This indicates that our proposal is robust to variations in grid field shape and to the precise decoding algorithm (*Figure 3C*). The scaling ratio $r$ may lie anywhere within a basin around the optimum at the cost of a small number of additional neurons. Such considerations also suggest that these coding schemes have the capacity to tolerate developmental noise: different animals could develop grid systems with slightly different scaling ratios, without suffering a large loss in efficiency. In two dimensions, the required neuron number will be no more than 5% of the minimum if the scale factor is within the range (1.43, 1.96) for the winner-take-all model and the range (1.28, 1.66) for the probabilistic model. These 'optimal intervals' are narrower than in the one-dimensional case and have substantial overlap.

In summary, for 2-d case, the theory predicts that (1) the ratios between adjacent scales should be a constant; (2) the optimal scaling constant is $\sqrt{e} \approx 1.65$ in a simple WTA decoding model, and it is $\approx 1.44$ in a probabilistic decoding model; (3) the predictions for the optimal grid field width depends on the specific decoding method, (4) The grid lattice should be a triangular lattice.

## Comparison to experiment

Our predictions agree with experiment (*Barry et al., 2007*; *Giocomo et al., 2011a*; *Stensola et al., 2012*) (see Reanalysis of grid data from previous studies, 'Materials and methods' for details of the data re-analysis). Specifically, *Barry et al. (2007)* (*Figure 4A*) reported the grid periodicities measured at three locations along the dorso–ventral axis of the MEC in rats and found ratios of ~1, ~1.7 and ~2.5 ≈ 1.6 × 1.6 relative to the smallest period (*Barry et al., 2007*). The ratios of adjacent scales reported in *Barry et al. (2007)* had a mean of 1.64 ± 0.09 (mean ± std. dev., $n = 6$), which almost precisely matches the mean scale factor of $\sqrt{e}$ predicted from the winner-take-all decoding model, and is also consistent with the probabilistic decoding model. In another study (*Krupic et al., 2012*), the scale ratio between the two smaller grid scales, measured by the ratio between the grid frequencies, is reported to be ~1.57 in one animal. Recent analysis based on a larger data set (*Stensola et al., 2012*) confirms the geometric progression of the grid scales in individual animals over four modules. The mean ratio between adjacent scales is 1.42 ± 0.17 (mean ± std. dev., $n = 24$) in that data set, accompanied by modest variability within and between animals. These measurements again match both our models (*Figure 4A*).

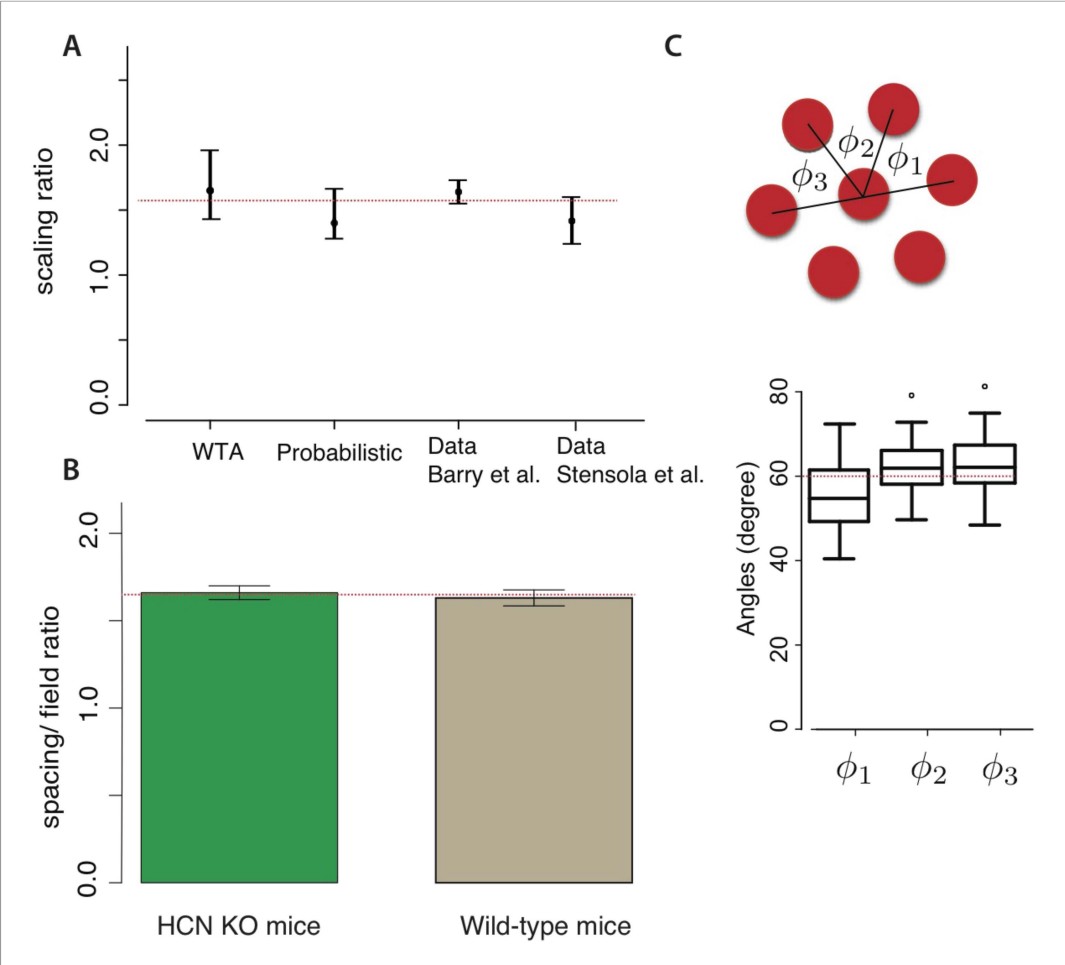

**Figure 4**. Comparison with experiment. (**A**) Our models predict grid scaling ratios that are consistent with experiment. 'WTA' (winner-take-all) and 'probabilistic' represent predictions from two decoding models; the dot is the scaling ratio minimizing the number of neurons, and the bars represent the interval within which the neuron number will be no more than 5% higher than the minimum. For the experimental data, the dot represents the mean measured scale ratio, and the error bars represent ± one standard deviation. Data were replotted from **Barry et al. (2007)**; **Stensola et al. (2012)**. The dashed red line shows a consensus value running through the two theoretical predictions and the two experimental datasets. (**B**) The mean ratio between grid periodicity ($\lambda_i$) and the diameter of grid fields ($l_i$) in mice (data from **Giocomo et al., 2011a**). Error bars indicate ± one S.E.M. For both wild-type mice and HCN knockouts (which have larger grid periodicities), the ratio is consistent with $\sqrt{e}$ (dashed red line). (**C**) The response lattice of grid cells in rats forms an equilateral triangular lattice with 60° angles between adjacent lattice edges (replotted from **Hafting et al., 2005**, $n = 45$ neurons from six rats). Dots represent outliers, as reported in **Hafting et al. (2005)**.

The optimal grid was triangular in both of our models, again matching measurements (**Figure 4C**) (**Hafting et al., 2005**; **Moser et al., 2008**; **Stensola et al., 2012**). However, the minimum in **Figure 3D** is relatively shallow—the contour lines indicating equally efficient grids are widely spaced near the minimum. This leads us hypothesize that the measured grid geometries will be modestly variable around the triangular lattice, as reported in **Stensola et al. (2012)**.

A recent study measured the ratio between grid periodicity and grid field size to be $1.63 \pm 0.035$ (mean ± S.E.M., $n = 48$) in wild-type mice (**Giocomo et al., 2011a**). This ratio was unchanged, $1.66 \pm 0.03$ (mean ± S.E.M., $n = 86$), in HCN1 knockout strains whose absolute grid periodicities increased relative to the wild type (**Giocomo et al., 2011a**). Such measurements are consistent with the prediction of the simple winner-take-all model, which predicts a ratio between grid period and grid field width of $\lambda_i/l_i = \sqrt{e} \approx 1.65$ (**Figure 4B**).

## Discussion

We have shown that a grid system with a discrete set of periodicities, as found in the entorhinal cortex, should use a common scale factor $r$ between modules to represent spatial location with the fewest neurons. In other words, the periods of grid modules should be organized in a geometric progression. In one dimension, this organization may be thought of intuitively as implementing a neural analog of a base-$b$ number system. Roughly, the largest scale localizes the animal into a coarse region of the environment and finer scales successively subdivide the region into $b$ 'bins'. For example, suppose that the largest scale has one firing field in the environment and that $b = 2$, so that subsequent scales subdivide this firing field into halves (*Figure 1B*). Then, keeping track of which half the animal occupies at each scale gives a binary encoding of location. This is just like a binary number system being used to encode a number representing the location. Our problem of minimizing neuron number while fixing resolution is analogous to minimizing the product of the number of digits and the number of decimal places (which we can term *complexity*) needed to represent a given range $R$ of integers in a base-$b$ number system. The complexity is approximately $C \sim b \log_b R$. What 'base' minimizes the complexity of the representation? We can compute this by evaluating the extremum $\partial C / \partial b = 0$ and find that the optimum is at $b = e$ (details in Optimizing a 'base-b' representation of one-dimensional space, 'Materials and methods'). Our full theory is a generalization of this simple fixed-base representational scheme for numbers to noisy neurons encoding two-dimensional location. It is remarkable that natural selection seems to have reached such efficient solutions for encoding location.

Our theory quantitatively predicted the ratios of adjacent scales within the variability tolerated by the models and by the data (*Figure 4*). Further tests of our theory are possible. For example, a direct generalization of our reasoning says that in n-dimensions the optimal ratio between grid scales for winner-take-all decoding is $\sqrt[n]{e}$ (as compared to $\sqrt{e}$ in two dimensions). The three-dimensional case is possibly relevant to the grid system in, for example, bats (*Yartsev et al., 2011*; *Yartsev and Ulanovsky, 2013*). Robustly, for any given decoding scheme, our theory would predict a smaller scaling ratio for 3d grids than for 2d grids. The packing density argument given above for two-dimensional lattice structure, when generalized to three dimensions, would predict a face center cubic lattice or hexagonal close packing, which share the highest packing density. Bats are known to have 2d grids when crawling on surfaces (*Yartsev et al., 2011*) and if they also have a 3d grid system when flying, similar to their place cell system (*Yartsev and Ulanovsky, 2013*), our predictions for three-dimensional grids can be directly tested. In general, the theory can be tested by comprehensive population recordings of grid cells along the dorso–ventral axis for animals moving in one-, two-, and three-dimensional environments.

Our theory also predicts a logarithmic relationship between the natural behavioral range and the number of grid modules. To estimate the number of modules, $m$, required for a given resolution $R_2$ via the approximate relationship $m = \log R_2 / \log \tilde{r}^2$. Assuming that the animal must be able to represent an environment of area $\sim (10 \text{ m})^2$ (e.g., *Davis et al., 1948*), with a positional accuracy on the scale of the rat's body size, $\sim (10 \text{ cm})^2$, we get a resolution of $R_2 \sim 10^4$. Together with the predicted two-dimensional scale factor $\tilde{r}$, this gives $m \approx 10$ as an order-of-magnitude estimate. Indeed, in *Stensola et al. (2012)*, 4–5 modules were discovered in recordings spanning up to 50% of the dorsoventral extent of MEC; extrapolation gives a total module number consistent with our estimate.

How many grid cells do we predict in total? Consider the simplest case where grid cells are independent encoders of position in two dimensions. Our likelihood analysis (details in Optimizing the grid system: probabilistic decoder, 'Materials and methods') gives the number of neurons as $N = mc (\lambda/\sigma)^2$, where $m$ is the number of modules and $c$ is constant. In detail, $c$ is determined by factors like the tuning curve shape of individual neurons and their firing rates, but broadly what matters is the typical number of spikes $K$ that a neuron emits during a sampling time, because this will control the precision with which location can be inferred from a single cell's response. General considerations (*Dayan and Abbott, 2001*) indicate that $c$ will be proportional to $1/K$. We can estimate that if a rat runs at $\sim 50$ cm/s and covers $\sim 1$ cm in a sampling time, then a grid cell firing at 10 Hz (*Stensola et al., 2012*) gives $K \sim 1/5$. Using our prediction that the number of modules will be $\sim 10$ and that $\lambda/\sigma \approx 5.3$ in the optimal grid (see Optimizing the grid system: probabilistic decoder, 'Materials and methods'), we get $N_{est} \approx 1400$. This estimate assumed independent neurons and that the decoder of the grid system will efficiently use all the information in every grid cell's response. This is unlikely to be the case. Given homogeneous noise

correlations within a grid module, which will arise naturally if grid cells are formed by an attractor mechanism, the required number of neurons could be an order of magnitude higher (*Sompolinsky et al., 2001*; *Averbeck et al., 2006*). (Noise correlation between grid cells was investigated in *Mathis et al. (2013)*; *Dunn et al. (2015)*—they found positive correlation between aligned grids of similar periods and some evidence for weak negative correlation for grids differing in phase.) Thus, in round numbers, we estimate that our theory requires something in the range of ~1400–14000 grid cells.

Are there so many grid cells in the MEC? In fact, we need this number of grid cells separately in layer II and layer III of the MEC since these regions likely maintain separate grid codes. (To see this, recall that layers II and III project largely to the dentate gyrus and CA1, respectively [*Steward and Scoville, 1976*; *Dolorfo and Amaral, 1998*], while the place map in CA1 survives lesions of the dentate input to CA1 via CA3 [*Brun et al., 2002*].) Physiological studies (*Sargolini et al., 2006*) have shown that only about 10% of the cells in MEC are layer II grid cells and another 10% are layer III grid cells. Cells that have weak responsiveness during spatial tasks are probably undersampled in such experiments and so the real proportion of grid cells is likely to be somewhat smaller. Other studies (*Mulders et al., 1997*) have shown that MEC has ~$10^5$ neurons. Thus, we can estimate that layer II and layer III each contain something in the range of 5000–10000 grid cells. This is well within the predicted theoretical range.

Our analysis assumed that the grid code is hierarchical, with large grids resolving the spatial ambiguity created by the multiple firing fields of the small grids that deliver precision of location. Recall that place cells are thought to provide one readout of the grid system. Anatomical evidence (*Van Strien et al., 2009*) shows that the projections from the mEC to the hippocampus are restricted along the dorso-ventral axis, so that a given place cell receives input from perhaps a quarter of the mEC. The data of *Stensola et al. (2012)* show additionally that the dorsal mEC is impoverished in large grid modules. If place cells were formed from grids via summation as in the model of (*Solstad et al., 2006*), the anatomy (*Van Strien et al., 2009*) and the hierarchical view of location coding that we have proposed would together predict that dorsal place cells should be revealed to have multiple place fields in large environments because their spatial ambiguities will not be fully resolved at larger scales. Preliminary evidence for such a multiplicity of dorsal place fields appears in *Fenton et al. (2008)*; *Rich et al. (2014)*. However, a naive model where place cells are sums of grid cells would also suggest that the multiple place fields would be arranged in an orderly, possibly periodic, manner. To the contrary, the data (*Fenton et al., 2008*; *Rich et al., 2014*) show that the multiple place fields of dorsal hippocampal cells are organized in a disorderly fashion. On the other hand, real grid fields show significant variability in period, orientation, and ellipticity even within a module (*Stensola et al., 2012*)—this variability would disorder any linearly summed place fields, changing the prediction of the naive model. We have not attempted to investigate this in detail because there is also significant evidence (summarized in *Bush et al., 2014*; *Sasaki et al., 2015*) that place cells are not formed and maintained via simple summation of grid cells alone, although they are influenced by them. It would be interesting for future work to integrate the accumulating information about the complex interplay between the hippocampus and the mEC to better understand the consequences of hierarchical grid organization for the hippocampal place system.

We assumed that the largest scales of grid modules should be roughly comparable to the behavioral range of the animal. This is consistent with the existing data on grid modules (*Stensola et al., 2012*) and with measurements in the largest environments tested so far (*Brun et al., 2008*) (periods at least as large as 10 m in an 18 m track). To accommodate very large environments, grids could either increase their scale (as reported at least transiently in *Barry et al., 2007*; *Stensola et al., 2012*) or could segment the environment into large sections (*Derdikman et al., 2009*; *Derdikman and Moser, 2010*) across which remapping occurs (*Fyhn et al., 2007*). These predictions can be tested in detail by exploring spatial coding in natural environments of behaviorally appropriate size and complexity. In fact, ethological studies have indicated a typical homing rate of a few tens of meters for rats with significant variation between strains (*Davis et al., 1948*; *Fitch, 1948*; *Stickel and Stickel, 1949*; *Slade and Swihart, 1983*; *Braun, 1985*). Our theory predicts that the period of the largest grid module and the number of modules will be correlated with homing range.

In our theory, we took the coverage factor *d* (the number of grid fields overlapping a given point in space) to be the same for each module. In fact, experimental measurements have not yet established whether this parameter is constant or varies between modules. How would a varying *d* affect our results? The answer depends on the dimensionality of the grid. In two dimensions, if neurons have

weakly correlated noise, modular variation of the coverage factor does not affect the optimal grid at all. This is because the coverage factor cancels out of all relevant formulae, a coincidence of two dimensions (see Optimizing the grid system: probabilistic decoder, 'Materials and methods', and p. 112 of *Dayan and Abbott, 2001*). In one and three dimensions, variation of $d$ between modules will have an effect on the optimal ratios between the variable modules. Thus, if the coverage factor is found to vary between grid modules for animals navigating one and three dimensions, our theory can be tested by comparing its predictions for the corresponding variations in grid scale factors. Similarly, even in two dimensions, if noise is correlated between grid cells, then variability in $d$ can affect our predicted scale factor. This provides another avenue for testing our theory.

The simple winner-take-all model assuming compact grid fields predicted a ratio of field width to grid period that matched measurements in both wild-type and HCN1 knockout mice (*Giocomo et al., 2011a*). Since the predicted grid field width is model dependent, the match with the simple WTA prediction might be providing a hint concerning the method the brain uses to read the grid code. Additional data on this ratio parameter drawn from multiple grid modules may serve to distinguish and select between potential decoding models for the grid system. The probabilistic model did not make a direct prediction about grid field width; it instead worked with the standard deviation $\sigma_i$ of the posterior $P(x|i)$. This parameter is predicted to be $\sigma_i = 0.19\lambda_i$ in two dimensions (see Optimizing the grid system: probabilistic decoder, 'Materials and methods'). This prediction could be tested behaviorally by comparing discrimination thresholds for location to the period of the smallest module. The standard deviation $\sigma_i$ can also be related to the noise, neural density and tuning curve shape in each module (*Dayan and Abbott, 2001*).

Previous work by *Fiete et al. (2008)* proposed that the grid system is organized to represent very large ranges in space by exploiting the incommensurability (i.e., lack of common rational factors) of different grid periods. As originally proposed, the grid scales in this scheme were not hierarchically organized (as we now know they are *Stensola et al., 2012*) but were of similar magnitude, and hence it was particularly important to suggest a scheme where a large spatial range could be represented using grids with small and similar periods. Using all the scales together (*Fiete et al., 2008*) argued that it is easy to generate ranges of representation that are much larger than necessary for behavior, and Sreenivasan and Fiete argued that the excess capacity could be used for error correction over distances relevant for behavior (*Sreenivasan and Fiete, 2011*). However, recent experiments tell us that there is a hierarchy of scales (*Stensola et al., 2012*) which should make the representation of behaviorally plausible range of 20–100 m easily accessible in the alternative hierarchical coding scheme that we have proposed. Nevertheless, we have checked that a grid coding scheme with the optimal scale ratio predicted by our theory can represent space over ranges larger than the largest grid period ('Range of location coding in a grid system', Appendix 1). However, to achieve this larger range, the number of neurons in each module will have to increase relative to the minimum in order to shrink the widths of the peaks in the likelihood function over position. It could be that animals sometimes exploit this excess capacity either for error correction or to avoid remapping over a range larger than the period of the largest grid. That said, experiments do tell us that remapping occurs readily over relatively small (meter length) scales at least for dorsal (small scale) place cells and grid cells (*Fyhn et al., 2007*) in tasks that involve spatial cues.

Our hierarchical grid scheme makes distinctive predictions relative to a non-hierarchical model for the effects of selective lesions of grid modules in the context of specific models where grid cells sum to make place cells (details in 'Predictions for the effects of lesions and for place cell activity', Appendix 1). In such a simple grid to place cell transformation, lesioning the modules with small periods will expand place field widths, while lesioning modules with large periods will lead to increased firing at locations outside the main place field, at scales set by the missing module. Similar effects are predicted for any simple decoder of a lesioned hierarchical grid system that has no other location related inputs—that is, animals with lesions to fine grid modules will show less precision in spatial behavior, while animals with lesions to large grid modules will confound well-separated locations. In contrast, in a non-hierarchical grid scheme with similar but incommensurate periods, lesions of any module lead to the appearance of multiple place fields at many scales for each place cell. Recent studies which ablated a large fraction of the mEC at all depths showed an increase in place field widths (*Hales et al., 2014*), as did the more focal lesions of *Ormond and McNaughton (2015)* along the dorso–ventral axis of the mEC. However, there are multiple challenges in interpreting these experiments. First, the data of *Stensola et al. (2012)* shows that there are modules

with both small and large periods at every depth along the mEC—the dorsal mEC is simply enriched in modules with large periods. So *Hales et al. (2014)*; *Ormond and McNaughton (2015)* are both removing modules that have both small and large periods. A simple linear transformation from a hierarchical grid to place cells would predict that removing large periods increases the number of place fields, but *Hales et al. (2014)* did not look for this effect while in *Ormond and McNaughton (2015)* the reported number of place fields decreases after lesions (including complete dirsruption of place fields of some cells). The underlying difficulty in interpretation is that while place cells might be summing up grid cells, there is evidence that they can be formed and maintained through mechanisms that may not critically involve the mEC at all (*Bush et al., 2014*; *Sasaki et al., 2015*). Thus, despite the interpretation given in *Kubie and Fox (2015)*; *Ormond and McNaughton (2015)* in favor of the partial validity of a linearly summed grid to place model, it is difficult for theory to make a definitive prediction for experiments until the inter-relation of the mEC and hippocampus is better understood.

   *Mathis et al. (2012a)* and *Mathis et al. (2012b)* studied the resolution and representational capacity of grid codes vs place codes. They found that grid codes have exponentially greater capacity to represent locations than place codes with the same number of neurons. Furthermore, *Mathis et al. (2012a)* predicted that in one dimension a geometric progression of grids that is self-similar at each scale minimizes the asymptotic error in recovering an animal's location given a fixed number of neurons. To arrive at these results the authors formulated a population coding model where independent Poisson neurons have periodic one-dimensional tuning curves. The responses of these model neurons were used to construct a maximum likelihood estimator of position, whose asymptotic estimation error was bounded in terms of the Fisher information—thus the resolution of the grid was defined in terms of the Fisher information of the neural population (which can, however, dramatically overestimate coding precision for neurons with multimodal tuning curves [*Bethge et al., 2002*]). Specializing to a grid system organized in a fixed number of modules, *Mathis et al. (2012a)* found an expression for the Fisher information that depended on the periods, populations, and tuning curve shapes in each module. Finally, the authors imposed a constraint that the scale ratio had to exceed some fixed value determined by a 'safety factor' (dependent on tuning curve shape and neural variability), in order reduce ambiguity in decoding position. With this formulation and assumptions, optimizing the Fisher information predicts geometric scaling of the grid in a regime where the scale factor is sufficiently large. The Fisher information approximation to position error in *Mathis et al. (2012a)* is only valid over a certain range of parameters. An ambiguity-avoidance constraint keeps the analysis within this range, but introduces two challenges for an optimization procedure: (i) the optimum depends on the details of the constraint, which was somewhat arbitrarily chosen and was dependent on the variability and tuning curve shape of grid cells, and (ii) the optimum turns out to saturate the constraint, so that for some choices of constraint the procedure is pushed right to the edge of where the Fisher information is a valid approximation at all, causing difficulties for the self-consistency of the procedure.

   Because of these limits on the Fisher information approximation, *Mathis et al. (2012a)* also measured decoding error directly through numerical studies. But here a complete optimization was not possible because there are too many inter-related parameters, a limitation of any numerical work. The authors then analyzed the dependence of the decoding error on the grid scale factor and found that, in their theory, the optimal scale factor depends on 'the number of neurons per module and peak firing rate' and, relatedly, on the 'tolerable level of error' during decoding (*Mathis et al., 2012a*). Note that decoding error was also studied in *Towse et al. (2014)* and those authors reported that the results did not depend strongly on the precise organization of scales across modules.

   In contrast to *Mathis et al. (2012a)*, we estimated decoding error directly by working with approximated forms of the likelihood function over position rather than by approximating decoding error in terms of the Fisher information. Conceptually, we can think of the winner-take-all analysis as effectively approximating the likelihood in terms of periodic boxcar functions; for the probabilistic analysis, we treat the likelihood as a periodic sum-of-Gaussians. Since at least scores of cells are being combined within modules, the Gaussian approximation to local likelihood peaks is valid, allowing us to circumvent detailed analysis of tuning curves and variability of individual neurons. These approximations allow analytical treatment of the optimization problem over a much wider parameter range without requiring arbitrary hand-imposed constraints. Our formulation of grid resolution then simply estimates the number of distinct regions that a fixed range can be divided into. We then fix this resolution as being behaviorally determined and minimize the number of required neurons while

allowing the periods of the modules, and, crucially, the number of modules, to vary to achieve the minimum.

All told, our simpler, and more intuitive, formulation of grid coding embodies very general considerations trading off precision and ambiguity with a sufficiently dense population of grid cells. The simplicity and generality of our setting allows us to make predictions for structural parameters of the grid system in different dimensions. These predictions—scaling ratios in 1, 2, and 3 dimensions; the ratio of grid period to grid field width; the number of expected modules; the shape of the optimal grid lattice; an estimate of the total expected number of grid cells—can be directly tested in experiments.

There is a long history in the study of sensory coding, especially vision, of identifying efficiency principles underlying neural circuits and codes starting with *Barlow (1961)*. Our results constitute evidence that such principles might also operate in the organization of cognitive circuits processing non-sensory variables. Furthermore, the existence of an efficiency argument for grid organization of spatial coding suggests that grid systems may be universal amongst the vertebrates, and not just a rodent specialization. In fact, there is evidence that humans (*Doeller et al., 2010*; *Jacobs et al., 2013*) and other primates (*Killian et al., 2012*) also have grid systems. We expect that our predicted scaling of the grid modules also holds in humans and other primates.

## Materials and methods

### Optimizing a 'base-b' representation of one-dimensional space

Suppose that we want to resolve location with a precision $l$ in a track of length $L$. In terms of the resolution $R = L/l$, we argued in the 'Discussion' that a 'base-b' hierarchical neural coding scheme will roughly require $N = b \log_b R$ neurons. To derive the optimal base (i.e., the base that minimizes the number of the neurons), we evaluate the extremum $\partial N/\partial b = 0$:

$$\partial N \Big/ \partial b = \frac{\partial(b \, \log_b R)}{\partial b} = \frac{\partial\left(\frac{b \, \ln R}{\ln b}\right)}{\partial b} = \ln R \, \frac{\ln b - 1}{(\ln b)^2}. \qquad (1)$$

Setting $\partial N/\partial b = 0$ gives $\ln b - 1 = 0$. Therefore, the number of neurons is extremized when $b = e$. It is easy to check that this is a minimum. Of course, the base of a number system is usually taken to be an integer, so the argument should be taken as motivating the more detailed treatment of neural representations of space above. Neurons are of course not constrained to organize the periodicity of their tuning curves in integer ratios.

### Optimizing the grid system: winner-take-all decoder

#### Deriving the optimal grid

We have seen that, for a winner-take-all decoder, the problem of deriving the optimal ratios of adjacent grid scales in one dimension is equivalent to minimizing the sum of a set of numbers ($N = d \sum_{i=1}^{m} r_i$) while fixing the product ($R_1 = \prod_{i=1}^{m} r_i$) to take the value $R$. Mathematically, it is equivalent to minimize $N$ while fixing $\ln R_1$. When $N$ is large, we can treat it as a continuous variable and use the method of Lagrange multipliers as follows. First, we construct the auxiliary function $H(r_1 \cdots r_m, \beta) = N - \beta \, (\ln R_1 - \ln R)$ and then extremize $H$ with respect to each $r_i$ and $\beta$. Extremizing with respect to $r_i$ gives

$$\frac{\partial H}{\partial r_i} = d - \frac{\beta}{r_i} = 0 \quad \Rightarrow \quad r_i = \frac{\beta}{d} \equiv r \, . \qquad (2)$$

Next, extremizing with respect to $\beta$ to implement the constraint on the resolution gives

$$\frac{\partial H}{\partial \beta} = \ln R_1 - \ln R = m \ln r - \ln R = 0 \quad \Rightarrow \quad r = R^{1/m}. \qquad (3)$$

Having thus implemented the constraint that $\ln R_1 = \ln R$, it follows that $H = N = dmR^{1/m}$. Alternatively, solving for $m$ in terms of $r$, we can write $H = d\,r\,(\ln R)/\ln r) = d\,r\,\log_r R$. It remains to minimize the number of cells $N$ with respect to $r$,

$$\frac{\partial H}{\partial r} = d \ \ln R \left[ \frac{1}{\ln r} - \left( \frac{1}{\ln r} \right)^2 \right] = 0 \quad \Rightarrow \quad \ln r = 1. \tag{4}$$

This is in turn implies our result

$$r = e, \tag{5}$$

for the optimal ratio between adjacent scales in a hierarchical, grid coding scheme for position in one dimension, using a winner-take-all decoder. In this argument, we employed the sleight of hand that $N$ and $m$ can be treated as continuous variables, which is approximately valid when $N$ is large. This condition obtains if the required resolution $R$ is large. A more careful argument is given below that preserves the integer character of $N$ and $m$.

### Integer $N$ and $m$

Above we used Lagrange multipliers to enforce the constraint on resolution and to bound the scale ratios to avoid ambiguity while minimizing the number of neurons required by a winner-take-all decoding model of grid systems. Here, we will carry out this minimization while recognizing that the number of neurons is an integer. First, consider the arithmetic mean–geometric mean inequality which states that, for a set of non-negative real numbers, $x_1, x_2,..., x_m$, the following holds:

$$(x_1 + x_2 + \ldots + x_m) \Big/ m \geq (x_1 x_2 \ldots x_m)^{1/m} , \tag{6}$$

with equality if and only if all the $x_i$'s are equal. Applying this inequality, it is easy to see that to minimize $\sum_{i=1}^{m} r_i$, all of the $r_i$ should be equal. We denote this common value as $r$, and we can write $r = R^{1/m}$.

Therefore, we have

$$N = d \sum_{i=1}^{m} r = m \ d \ R^{1/m}. \tag{7}$$

Suppose $R = e^{z + \epsilon}$, where $z$ is an integer, and $\epsilon \in [0, 1)$. By taking the first derivative of N with respect to m, and setting it to zero, we find that $N$ is minimized when $m = z + \epsilon$. However, since $m$ is an integer the minimum will be achieved either at $m = z$ or $m = z + 1$. (Here, we used the fact $mR^{1/m}$ is monotonically increasing between 0 and $z + \epsilon$ and is monotonically decreasing between $z + \epsilon$ and $\infty$.) Thus, minimizing N requires either

$$r = (e^{z + \epsilon})^{\frac{1}{z}} = e^{\frac{z+\epsilon}{z}} \quad \text{or} \quad r = (e^{z + \epsilon})^{\frac{1}{z+1}} = e^{\frac{z+\epsilon}{z+1}} . \tag{8}$$

In either case, when $z$ is large (and therefore $R$, $N$ and $m$ are large), $r \to e$. This shows that when the resolution $R$ is sufficiently large, the total number of neurons $N$ is minimized when $r_i \approx e$ for all $i$.

### Optimal winner-take-all grids: general formulation

As described in the above, we wish to choose the grid system parameters $\{\lambda_i, l_i\}$, $1 \leq i \leq m$, as well as the number of scales $m$, to minimize neuron number:

$$N = d \sum_{i=1}^{m} \frac{\lambda_i}{l_i}, \tag{9}$$

where $d$ is the fixed coverage factor in each module, while constraining the positional accuracy of the grid system and the range of representation. We can take the positional accuracy to be proportional to the grid field width of the smallest module. This gives

$$c_1 \ l_m = A. \tag{10}$$

To give a sufficiently large range of representation in our hierarchical scheme we will require that

$$\lambda_1 \geq L. \tag{11}$$

Following the main text, to eliminate ambiguity at each scale we need that

$$\lambda_{i+1} \geq c_2 \, l_i, \tag{12}$$

where $c_2$ depends on the tuning curve shape and coverage factor (written as $f(d)$ above).

We will first fix $m$ and solve for the remaining parameters, then optimize over $m$ in a subsequent step. Optimization problems subject to inequality constraints may be solved by the method of Karush-Kuhn-Tucker (KKT) conditions (**Kuhn and Tucker, 1951**). We first form the Lagrange function,

$$\mathscr{L} = d \sum_i \frac{\lambda_i}{l_i} + \alpha(c_1 \, l_m - A) - \beta_0(\lambda_1 - L) - \sum_{i=1}^{K-1} \beta_i(\lambda_{i+1} - c_2 \, l_i). \tag{13}$$

The KKT conditions include that the gradient of $\mathscr{L}$ with respect to $\{\lambda_i,...,l_i\}$ vanish,

$$\frac{\partial \mathscr{L}}{\partial l_m} = c_1 \alpha - d \frac{\lambda_m}{l_m^2} = 0, \tag{14}$$

$$\frac{\partial \mathscr{L}}{\partial l_i} = c_2 \beta_i - d \frac{\lambda_i}{l_i^2} = 0 \quad i < m, \tag{15}$$

$$\frac{\partial \mathscr{L}}{\partial \lambda_i} = \frac{d}{l_i} - \beta_{i-1} = 0, \tag{16}$$

together with the 'complementary slackness' conditions,

$$\beta_0(\lambda_1 - L) = 0, \tag{17}$$

$$\beta_i(\lambda_{i+1} - c_2 \, l_i) = 0. \tag{18}$$

From **Equations 15, 16**, we obtain:

$$\beta_i = \frac{d}{c} \frac{\lambda_i}{l_i^2} = \frac{d}{l_{i+1}}. \tag{19}$$

It follows that $\beta_i \neq 0$, and so the complementary slackness conditions give:

$$\lambda_1 = L, \tag{20}$$

$$\lambda_i = c_2 \, l_{i-1}. \tag{21}$$

Substituting this result into **Equation 19** yields,

$$r_i \equiv \frac{l_{i-1}}{l_i} = \frac{l_i}{l_{i+1}} = r_{i+1}, \tag{22}$$

that is, the scale factor $r$ is the same for all modules. Once we obtain a value for $r$, **Equations 20–22** yield values for all $\lambda_i$ and $l_i$. Since the resolution constraint may now be rewritten,

$$A = c_1 \, r^{-m} L, \tag{23}$$

we have $m = \ln(c_1 L/A)/\ln r$. Therefore, $r$ determines $m$ and so minimizing $N$ over $m$ is equivalent to minimizing over $r$. Expressing $N$ entirely in terms of $r$ gives,

$$N = d \, c_2 \, \ln(c_1 L/A) \frac{\ln r}{r}. \tag{24}$$

Optimizing with respect to $r$ gives the result $r = e$, independent of $d$, $c_1$, $c_2$, $L$, and $R$.

## Optimizing the grid system: probabilistic decoder

Consider a probabilistic decoder of the grid system that pools all the information available in the population of neurons in each module by forming the posterior distribution over position given the neural activity. In this general setting, we assume that the firing of different grid cells is weakly

correlated, that noise is homogeneous, and that the tuning curves in each module $i$ provide dense, uniform, coverage of the interval $\lambda_i$. With these assumptions, we will first consider the one-dimensional case, and then analyze the two-dimensional case by analogy.

## One-dimensional grids

With the above assumptions, the likelihood of the animal's position, given the activity of grid cells in module $i$, $P(x|i)$, can be approximated as a series of Gaussian bumps of standard deviation $\sigma_i$ spaced at the period $\lambda_i$ (**Dayan and Abbott, 2001**). As defined in 'Results', the number of cells ($n_i$) in the $i$th module, is expressed in terms of the period ($\lambda_i$), the grid field width ($l_i$) and a 'coverage factor' $d$ representing the cell density as $n_i = d\lambda_i/l_i$. The coverage factor $d$ will control the relation between the grid field width $l_i$ and the standard deviation $\sigma_i$ of the local peaks in the likelihood function of location. If $d$ is larger, $\sigma_i$ will be narrower since we can accumulate evidence from a denser population of neurons. The ratio $\frac{l_i}{\sigma_i}$ in general will be a monotonic function of the coverage factor $d$, which we will write as $\frac{l_i}{\sigma_i} = g(d)$. In the special case where the grid cells have independent noise $g(d) \propto \sqrt{d}$, so that $\sigma_i/l_i \propto 1/\sqrt{d}$—that is, the precision increases as the inverse square root of the cell density, as expected because the relevant parameter is the number of cells within one grid field rather than the total number of cells. Note that this does *not* imply an inverse square root relation between the *number* of cells $n_i$ and $\sigma_i$, because $n_i$ is also proportional to the period $\lambda_i$, and in our formulation the density $d$ is fixed while $\lambda_i$ can be varied. Note also that if the neurons have correlated noise, $g(d)$ may scale substantially slower than $\sqrt{d}$ (**Britten et al., 1992**; **Zohary et al., 1994**; **Sompolinsky et al., 2001**). Putting all of these statements together, we have, in general, $n_i = \frac{d}{g(d)}\frac{\lambda_i}{\sigma_i}$. Assuming that the coverage factor $d$ is the same across modules, we can simplify the notation and write $n_i = c\frac{\lambda_i}{\sigma_i}$, where $c = d/g(d)$ is a constant. (Again, for independent noise $\sigma_i \propto 1/d$ as expected—see above—and this does *not* imply a similar relationship to the number of cells $n_i$ as one might have naively assumed.) In sum, we can write the total number of cells in a grid system with $m$ modules as $N = \sum_{i=i}^{m} n_i = c \sum_{i=1}^{m} \frac{\lambda_i}{\sigma_i}$.

The likelihood of position derived from each module can be combined to give an overall probability distribution over location. Let $Q_i(x)$ be the likelihood obtained by combining modules 1 (the largest period) through $i$. Assuming that the different modules have independent noise, we can compute $Q_i(x)$ from the module likelihoods as $Q_i(x) \propto \prod_{j=1}^{i} P(x|j)$. We will take the prior probability over locations be uniform here so that this combined likelihood is equivalent to the Bayesian posterior distribution over location. The likelihoods from different scales have different periodicities, so multiplying them against each other will tend to suppress all peaks except the central one, which is aligned across scales. We may thus approximate $Q_i(x)$ by single Gaussians whose standard deviations we will denote as $\delta_i$. (The validity of this approximation is taken up in further detail below.)

Since $Q_i(x) \propto Q_{i-1}(x)P(x|i)$, $\delta_i$ is determined by $\delta_{i-1}$, $\lambda_i$ and $\sigma_i$. These all have dimensions of length. Dimensional analysis (**Rayleigh, 1896**) therefore says that, without loss of generality, the ratio $\delta_i/\delta_{i-1}$ can be written as a dimensionless function of any two cross-ratios of these parameters. It will prove useful to use this freedom to write $\delta_i = \delta_{i-1}/\rho\left(\frac{\lambda_i}{\sigma_i}, \frac{\sigma_i}{\delta_{i-1}}\right)$. The standard error in decoding the animal's position after combining information from all the grid modules will be proportional to $\delta_m$, the standard deviation of $Q_m$. We can iterate our expression for $\delta_i$ in terms of $\delta_{i-1}$ to write $\delta_m = \delta_0/\prod_{i=1}^{m} \rho_i$, where $\delta_0$ is the uncertainty in location without using any grid responses at all. (We are abbreviating $\rho_i = \rho(\lambda_i/\sigma_i, \sigma_i/\delta_{i-1})$). In the present probabilistic context, we can view $\delta_0$ as the standard deviation of the a priori distribution over position before the grid system is consulted, but it will turn out that the precise value or meaning of $\delta_0$ is unimportant. We assume a behavioral requirement that fixes $\delta_m$ and thus the resolution of the grid, and that $\delta_0$ is likewise fixed by the behavioral range. Thus, there is a constraint on the product $\prod_i \rho_i$.

Putting everything together, we wish to minimize $N = c \sum_{i=1}^{m} \frac{\lambda_i}{\sigma_i}$ subject to the constraint that $R = \prod_{i=1}^{m} \rho_i$, where $\rho_i$ is a function of $\lambda_i/\sigma_i$ and $\sigma_i/\delta_{i-1}$. Given the formula for $\rho_i$ derived in the next section, this can be carried out numerically. To understand the optimum, it is helpful to observe that the problem has a symmetry under permutations of $i$. So we can guess that in the optimum all the $\lambda_i/\sigma_i$, $\sigma_i/\delta_{i-1}$ and $\rho_i$ will be equal to a fixed $\lambda/\sigma$, $\sigma/\delta$, and $\rho$. We can look for a solution with this symmetry and then check that it is an optimum. First, using the symmetry, we write $N = cm(\lambda/\sigma)$ and $R = \rho^m$. It follows that $N = c(1/\ln\rho)(\lambda/\sigma)$ and we want to minimize it with respect to $\lambda/\sigma$ and $\sigma/\delta$. Now, $\rho(\lambda/\sigma, \sigma/\delta)$ is a complicated function of its arguments (**Equation 30**) which has a maximum value as a function of $\sigma/\delta$ for any fixed $\lambda/\sigma$. To minimize $N$ at fixed $\lambda/\sigma$, we should maximize $\rho$ with respect to $\sigma/\delta$ (**Figure 5**).

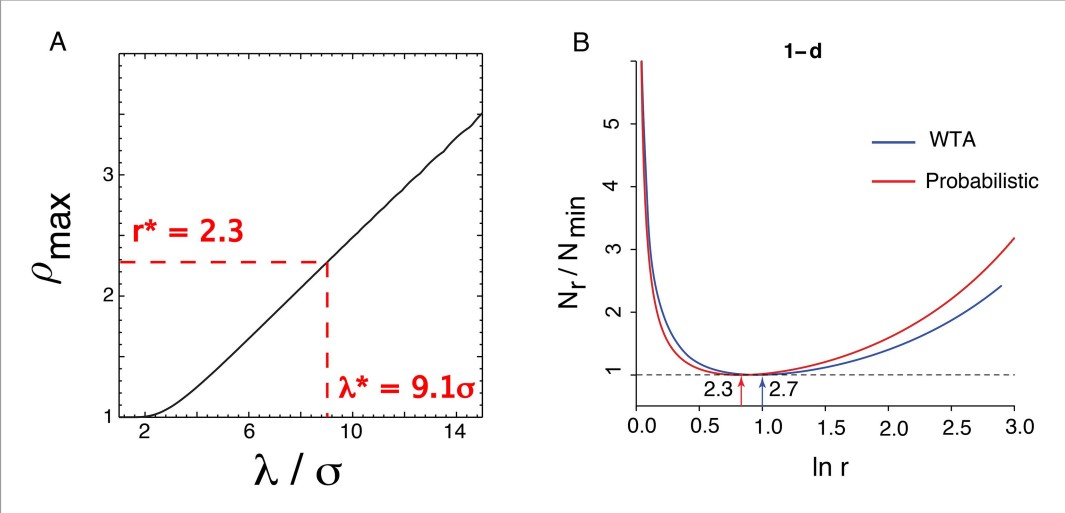

**Figure 5**. Optimizing the one-dimensional grid system. (**A**) $\rho_{max} \equiv \max_{\sigma/\delta} \rho\left(\frac{\lambda}{\sigma}, \frac{\sigma}{\delta}\right)$ is the scale factor after optimizing $N$ over $\sigma/\delta$. The values $r^*$ and $\lambda^*$ are the values chosen by the complete optimization procedure. (**B**) The optimal ratio $r$ between adjacent scales in a hierarchical grid system in one dimension for a simple winner-take-all decoding model (blue, WTA) and a probabilistic decoder (red). Here, $N_r$ is the number of neurons required to represent space with resolution $R$ given a scaling ratio $r$, and $N_{min}$ is the number of neurons required at the optimum. In both models, the ratio $N_r/N_{min}$ is independent of resolution, $R$. For the winner-take-all model, $N_r \propto r/\ln r$, while the curve for the probabilistic model is derived numerically (mathematical details in Optimizing the grid system: probabilistic decoder, 'Materials and methods'). The winner-take-all model predicts $r = e \approx 2.7$, while the probabilistic decoder predicts $r \approx 2.3$. The minima of the two curves lie within each others' shallow basins.

Given this $\rho_{max}$, we can minimize $N = c(\lambda/\sigma)/\ln \rho_{max}(\lambda/\sigma)$ with respect to $\lambda/\sigma$, and then plug back in to find the optimal $\rho$. It turns out to be $\rho^*_{max} = 2.3$.

In fact, $\rho$ is equal to the scale factor of the grid: $\rho_i = r_i = \lambda_i/\lambda_{i+1}$. To see this, we have to express $\rho_i$ in terms of the parameters $\lambda_i/\sigma_i$ and $\sigma_i/\delta_{i-1}$: $\rho_i = \frac{\delta_{i-1}}{\delta_i} = \frac{\delta_{i-1}}{\sigma_i} \frac{\sigma_i}{\lambda_i} \frac{\lambda_i}{\lambda_{i+1}} \frac{\lambda_{i+1}}{\sigma_{i+1}} \frac{\sigma_{i+1}}{\delta_i}$. Since the factors $\sigma_i/\delta_{i-1}$ and $\lambda_i/\sigma_i$ are independent of $i$, they cancel in the product and we are left with $\rho_i = \lambda_i/\lambda_{i+1}$.

Thus, the probabilistic decoder predicts an optimal scale factor $r^* = 2.3$ in one dimension. This is similar to, but somewhat different than, the winner-take-all result $r^* = e = 2.7$ (**Figure 5**). At a technical level, the difference arises because the function $\rho_{max}(\lambda/\sigma)$ is effectively $\rho_{max} \propto \frac{\lambda}{\sigma}$ in the winner-take-all analysis, but in the probabilistic case, it is more nearly a linear function with a positive offset $\rho \approx \alpha^{-1}\left(\frac{\lambda}{\sigma} + \beta\right)$. Conceptually, the optimal probabilistic scale factor is smaller in order to suppress side lobes that can arise in the combined likelihood across modules (**Figure 2**). Such side lobes were absent in the winner-take-all analysis. The optimization also predicts $\lambda^* = 9.1\sigma$. This relation between the period and standard deviation at each scale could be converted into a relation between grid period and grid field width given specific measurements of tuning curves, noise levels, and cell density in each module. For example, if neurons within a module have independent noise, then general population coding considerations (**Dayan and Abbott, 2001**) show that $\sigma = \beta d^{-1/2} l$, where $l$ is a measure of grid field width, $d$ is the density of neurons in a module, and $\beta$ is a dimensionless number that depends on noise (given the integration time) and tuning curve shape.and tuning curve shape.

## Two-dimensional grids

A similar probabilistic analysis can be carried out for two-dimensional grid fields. The posteriors $P(x|i)$ become two-dimensional sums-of-Gaussians, with the centers of the Gaussians laid out on the vertices of the grid. $Q_i(x)$ is then similarly approximated by a two-dimensional Gaussian. Generalizing from the one-dimensional case, the number of cells in module $i$ is given by $n_i = d(\lambda_i/l_i)^2$, where $d$ is density of grid fields. As in one dimension, increasing the density $d$ will decrease the standard deviation $\sigma_i$ of the local bumps in the posterior $P(x|i)$—that is, $l_i/\sigma_i = g(d)$, where $g$ is an increasing function of $d$. In the special case where the neurons have independent noise, $g(d) \propto d$ so that the precision measured by the standard deviation $\sigma_i$ decreases as the inverse square root of $d$. Putting all of these statements

together, we have, in general, $n_i = \frac{d}{g(d)^2}\left(\frac{\lambda_i}{\sigma_i}\right)^2$. In the special case where noise is independent so that $g(d) \propto d$, the density $d$ cancels out in this expression, and in this case, or when the density $d$ is the same across modules, we can write $n_i = c\left(\frac{\lambda_i}{\sigma_i}\right)$, where $c$ is just a constant. Redoing the optimization analysis from the one-dimensional case, the form of the function $\rho$ changes (Calculating $\rho\left(\frac{\lambda}{\sigma}, \frac{\sigma}{\delta}\right)$, 'Materials and methods'), but the logic of the above derivation is otherwise unaltered. In the optimal grid, we find that $\lambda^* \approx 5.3\sigma$ (or equivalently $\sigma \approx 0.19\lambda^*$).

## Calculating $\rho\left(\dfrac{\lambda}{\sigma}, \dfrac{\sigma}{\delta}\right)$

Above, we argued that the function $\rho\left(\frac{\lambda}{\sigma}, \frac{\sigma}{\delta}\right)$ can be computed by approximating the posterior distribution of the animal's position given the activity in module $i$, $P(x|i)$, as a periodic sum-of-Gaussians:

$$P(x \mid i) = \frac{1}{2K+1} \sum_{n=-K}^{K} \frac{1}{\sqrt{2\pi\sigma_i^2}} e^{-\frac{1}{2\sigma_i^2}(x - n\lambda_i)^2}, \tag{25}$$

where $K$ is assumed large. We further approximate the posterior given the activity of *all* modules coarser than $\lambda_i$ by a Gaussian with standard deviation $\delta_{i-1}$:

$$Q_{i-1}(x) = \frac{1}{\sqrt{2\pi\delta_{i-1}^2}} e^{-x^2/2\delta_{i-1}^2}. \tag{26}$$

(We are assuming here that the animal is really located at $x = 0$ and that the distributions $P(x|i)$ for each $i$ have one peak at this location.) Assuming noise independence across scales, it then follows that $Q_i(x) = \frac{P(x \mid i)Q_{i-1}(x)}{\int dx\, P(x \mid i)Q_{i-1}(x)}$. Then $\rho(\lambda_i/\sigma_i, \sigma_i/\delta_{i-1})$ is given by $\delta_{i-1}/\delta_i$, where $\delta_i$ is the standard deviation of $Q_i$. We therefore must calculate $Q_i(x)$ and its variance in order to obtain $\rho$. After some algebraic manipulation, we find,

$$Q_i(x) = \sum_{n=-K}^{K} \pi_n \frac{1}{\sqrt{2\pi\Sigma^2}} e^{-(x-\mu_n)^2/2\Sigma^2}, \tag{27}$$

where $\Sigma^2 = \left(\sigma_i^{-2} + \delta_{i-1}^{-2}\right)^{-1}$, $\mu_n = \left(\frac{\Sigma}{\sigma_i}\right)^2 \lambda_i\, n$, and

$$\pi_n = \frac{1}{Z} e^{-n^2\lambda_i^2/2\left(\sigma_i^2+\delta_{i-1}^2\right)}. \tag{28}$$

$Z$ is a normalization factor enforcing $\sum_n \pi_n = 1$. $Q_i$ is thus a mixture-of-Gaussians, seemingly contradicting our approximation that all the $Q$ are Gaussian. However, if the secondary peaks of $P(x|i)$ are well into the tails of $Q_{i-1}(x)$, then they will be suppressed (quantitatively, if $\lambda_i^2 \gg \sigma_i^2 + \delta_{i-1}^2$, then $\pi_n \ll \pi_0$ for $|n| \geq 1$), so that our assumed Gaussian form for $Q$ holds to a good approximation. In particular, at the values of $\lambda$, $\sigma$ and $\delta$ selected by the optimization procedure described above, $\pi_1 = 1.3 \times 10^{-3}\pi_0$. So our approximation is self-consistent.

Next, we find the variance $\delta_i^2$:

$$\delta_i^2 = \langle x^2 \rangle_{Q_i},$$

$$= \sum_n \pi_n \left(\Sigma^2 + \mu_n^2\right),$$

$$= \Sigma^2 \left(1 + \left(\frac{\Sigma}{\sigma_i}\right)^2 \left(\frac{\lambda_i}{\sigma_i}\right)^2 \sum_n n^2 \pi_n\right),$$

$$= \delta_{i-1}^2 \left(1 + \frac{\delta_{i-1}^2}{\sigma_i^2}\right)^{-1} \left(1 + \left(\frac{\Sigma}{\sigma_i}\right)^2 \left(\frac{\lambda_i}{\sigma_i}\right)^2 \sum_n n^2 \pi_n\right). \tag{29}$$

We can finally read off $\rho\left(\frac{\lambda_i}{\sigma_i}, \frac{\sigma_i}{\delta_{i-1}}\right)$ as the ratio $\delta_{i-1}/\delta_i$:

$$\rho\left(\frac{\lambda_i}{\sigma_i}, \frac{\sigma_i}{\delta_{i-1}}\right) = \left(1 + \frac{\delta_{i-1}^2}{\sigma_i^2}\right)^{1/2} \left(1 + \left(1 + \frac{\sigma_i^2}{\delta_{i-1}^2}\right)^{-1} \left(\frac{\lambda_i}{\sigma_i}\right)^2 \sum_n n^2 \pi_n\right)^{-1/2}. \qquad (30)$$

For the calculations reported in the text, we took $K = 500$.

We explained above that we should maximize $\rho$ over $\frac{\sigma}{\delta}$, while sholding $\frac{\lambda}{\sigma}$ fixed. The first factor in **Equation 30** increases monotonically with decreasing $\frac{\sigma}{\delta}$; however, $\sum_n n^2 \pi_n$ also increases and this has the effect of reducing $\rho$. The optimal $\frac{\sigma}{\delta}$ is thus controlled by a trade-off between these factors. The first factor is related to the increasing precision given by narrowing the central peak of $P(x|i)$, while the second factor describes the ambiguity from multiple peaks.

## Generalization to two-dimensional grids

The derivation can be repeated in the two-dimensional case. We take $P(x|i)$ to be a sum-of-Gaussians with peaks centered on the vertices of a regular lattice generated by the vectors $(\lambda_i \overrightarrow{u}, \lambda_i \overrightarrow{v})$. We also define $\delta_i^2 \equiv \frac{1}{2}\langle|x|^2\rangle_{Q_i}$. The factor of 1/2 ensures that the variance so defined is measured as an average over the two dimensions of space. The derivation is otherwise parallel to the above, and the result is,

$$\rho_2\left(\frac{\lambda_i}{\sigma_i}, \frac{\sigma_i}{\delta_{i-1}}\right) = \left(1 + \frac{\delta_{i-1}^2}{\sigma_i^2}\right)^{1/2} \left(1 + \frac{1}{2}\left(1 + \frac{\sigma_i^2}{\delta_{i-1}^2}\right)^{-1} \left(\frac{\lambda_i}{\sigma_i}\right)^2 \sum_{n,m} \left|n\overrightarrow{u} + m\overrightarrow{v}\right|^2 \pi_{n,m}\right)^{-1/2},$$

$$(31)$$

where $\pi_{n,m} = \frac{1}{Z}e^{-|n\overrightarrow{u} + m\overrightarrow{v}|^2 \lambda_i^2 / 2(\sigma_i^2 + \delta_{i-1}^2)}$.

## Reanalysis of grid data from previous studies

We reanalyzed the data from *Barry et al. (2007)* and *Stensola et al. (2012)* in order to get the mean and the variance of the ratio of adjacent grid scales. For *Barry et al. (2007)*, we first read the raw data from Figure 3B of their paper using the software GraphClick, which allows retrieval of the original (x,y)-coordinates from the image. This gave the scales of grid cells recorded from six different rats. For each animal, we grouped the grids that had similar periodicities (i.e., differed by less than 20%) and calculated the mean periodicity for each group. We defined this mean periodicity as the scale of each group. For four out of six rats, there were two scales in the data. For one out six rats, there were three grid scales. For the remaining rat, only one scale was obtained as only one cell was recorded from that rat. We excluded this rat from further analysis. We then calculated the ratio between adjacent grid scales, resulting in 6 ratios from five rats. The mean and variance of the ratio were 1.64 and 0.09, respectively ($n = 6$).

For *Stensola et al. (2012)*, we first read in the data using GraphClick from Figure 5D of their paper. This gave the scale ratios between different grids for 16 different rats. We then pooled all the ratios together and calculated the mean and variance. The mean and variance of the ratio were 1.42 and 0.17, respectively ($n = 24$).

*Giocomo et al. (2011a)* reported the ratios between the grid period and the *radius* of grid field (measured as the radius of the circle around the center field of the autocorrelation map of the grid cells) to be 3.26 ± 0.07 and 3.32 ± 0.06 for Wild-type and HCN KO mice, respectively. We halved these measurements to the ratios between grid period and the *diameter* of the grid field to facilitate the comparison to our theoretical predictions. The results are plotted in a bar graph (*Figure 4B*).

Finally, in *Figure 4C*, we replotted Figure 1C from *Hafting et al. (2005)* by reading in the data using GraphClick and then translating that information back into a plot.

## Acknowledgements

NSF grants PHY-1058202, EF-0928048, PHY-1066293, and PHY11-25915 supported this work, which was completed at the Aspen Center for Physics and the Kavli Institute for Theoretical Physics. VB was also supported by the Fondation Pierre Gilles de Gennes. JP was supported by the C.V. Starr Foundation. XW conceived of the project and developed the winner-take-all framework with VB. JSP developed the probabilistic framework and two-dimensional grid optimization. VB and XW carried out simulated lesion studies. XW, JSP, and VB wrote the article.

## Additional information

### Funding

| Funder | Grant reference | Author |
|---|---|---|
| National Science Foundation (NSF) | PHY-1058202 | Xue-Xin Wei, Jason Prentice, Vijay Balasubramanian |
| PSL Research University Paris | Fondation Pierre-Gilles de Gennes | Vijay Balasubramanian |
| The Starr Foundation | | Jason Prentice |
| National Science Foundation (NSF) | PHY-1066293 | Vijay Balasubramanian |
| National Science Foundation (NSF) | EF-0928048 | Xue-Xin Wei, Jason Prentice, Vijay Balasubramanian |
| National Science Foundation (NSF) | PHY-1125915 | Vijay Balasubramanian |

The funders had no role in study design, data collection and interpretation, or the decision to submit the work for publication.

### Author contributions

X-XW, JP, VB, Contributed to the conception and design of the theory, to the analysis and interpretation of data, and to the writing of the article, Conception and design, Analysis and interpretation of data, Drafting or revising the article

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

## Appendix 1

### Range of location coding in a grid system

The main text describes hierarchical grid coding schemes where the larger periods resolve ambiguity and smaller periods give precision in location coding. We took the largest grid period to be comparable to the behavioral range. In fact, if the periods $\lambda_i$ of the different modules are incommensurate with each other (i.e., they do not share common integer factors), it should be possible to resolve location over ranges larger than the largest grid period (*Fiete et al., 2008*; *Sreenivasan and Fiete, 2011*). The grid schemes that we predict share this virtue since they predict scale ratios that are not simple rational numbers. However, the precise maximum range will also depend on the widths of the grid fields $l_i$ relative to the period and on the number of grid cells $n_i$ in each module. In the probabilistic decoding scheme described in the main text, these parameters determine the standard deviation $\sigma_i$ of the periodic peaks in the likelihood of position given the activity in module $i$. The full range of unambiguous location representation depends on the ratios $\lambda_i/\sigma_i$. Increasing this ratio will tend to increase the range of unambiguous representation, but at the cost of increasing the number of cells in each module.

To illustrate, consider a one-dimensional grid system with four modules with a ratio of 2.7 between adjacent scales (this is close to the optimal ratio predicted by our analysis). Suppose the animal's true location is at 0. We can calculate the overall probability of the animal's location by multiplying together the likelihood functions resulting from activity in each individual module (see main text for details). We will examine the extent to which location can be decoded unambiguously over a range $(-3\lambda_{max}, 3\lambda_{max})$ where $\lambda_{max}$ is the larges period. When $\lambda_i/\sigma_i$ is close to the value of 9.1 predicted by the probabilistic analysis in Optimizing the grid system: probabilistic decoder, 'Materials and methods', the overall likelihood shows substantial ambiguity over this range because of secondary peaks in the likelihood distribution (**Appendix figure 1A**). As $\lambda_i/\sigma_i$ increases (requiring more neurons in each module), these secondary peaks decrease in amplitude. In **Appendix figure 1B**, we show that when $\lambda_i/\sigma_i = 30$, the 4-module grid system can represent location at least within the range $(-3\lambda_{max}, 3\lambda_{max})$.

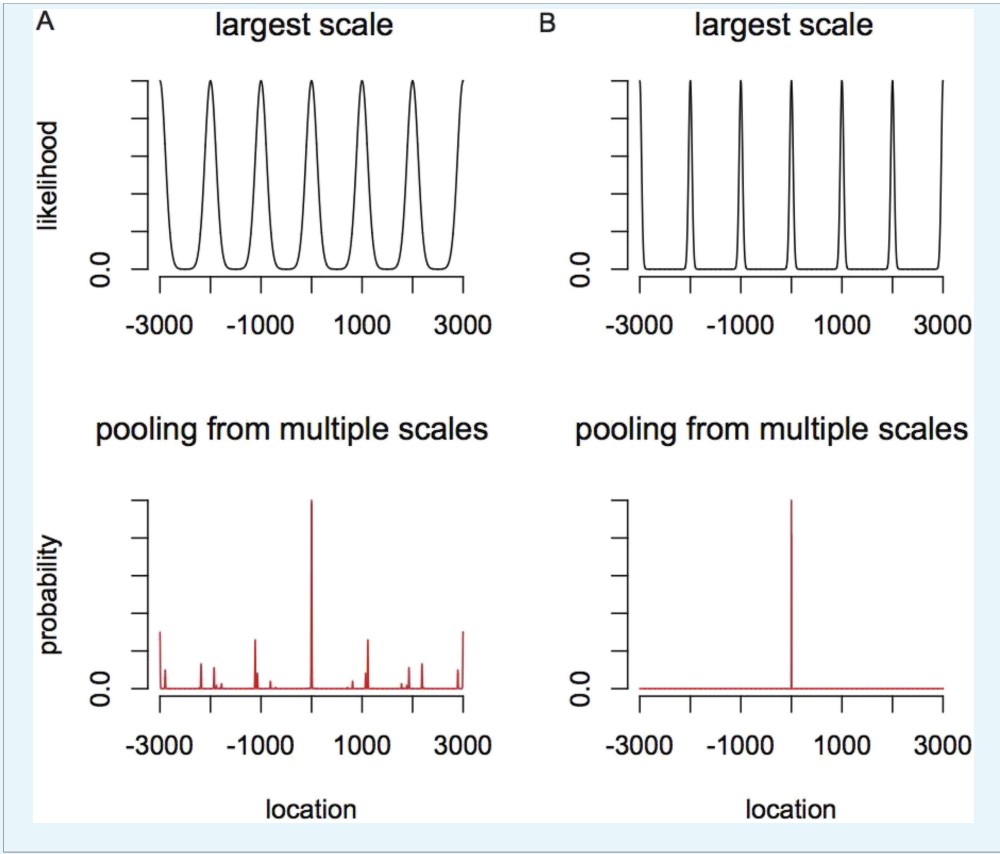

**Appendix figure 1**. Encoding range can exceed the period of the largest grid module at a cost in the number of neurons. Assume that the animal is located is at 0. (**A**) Top, the likelihood resulting from the largest grid module, where the standard deviation of the Gaussian peaks is $\frac{1}{9.1}$ of the grid period ($\lambda_{max} = 1000$). Bottom, the inferred distribution over location after pooling over 4-grid modules related by a scale factor of 2.7. As shown, this 4-module grid system shows ambiguities in location coding outside the range $[\lambda_{max}, \lambda_{max}]$. (**B**) Top, the likelihood resulting from the largest grid module, where the standard deviation of the Gaussian peaks is $\frac{1}{30}$ of the grid period ($\lambda_{max} = 1000$). Bottom, the inferred distribution over location after pooling over four grid modules related by a scale factor of 2.7. As shown, this 4-module grid system provides a good representation over a range of at least $[-3000, 3000] = [-3\lambda_{max}, 3\lambda_{max}]$.

If there is a biological limitation to the largest period possible in a grid system, and if the organism must represent very large ranges without grid remapping, it may prove beneficial to add neurons to expand range. Analyzing this trade-off requires knowledge of the range, biophysical limits on grid periods, and the degree of ambiguity (the maximum heights of secondary peaks in the probability of position) that can be behaviorally tolerated. This information is not currently available for any species, and so we do not attempt the analysis.

## Predictions for the effects of lesions and for place cell activity

In the grid coding scheme that we propose there is a hierarchy of grid periods governed by a geometric progression. The alternative schemes of *Fiete et al. (2008)*; *Sreenivasan and Fiete (2011)* are designed to produce a large range of representation from grids with *similar* periods. These two alternatives make very different predictions for the effects of lesions in the entorhinal cortex on location coding. In a hierarchical scheme, losing a grid module produces location ambiguities that increase in size with the period of the missing module. In the alternative scheme of *Fiete et al. (2008)*; *Sreenivasan and Fiete (2011)* lesions of a module produce periodic ambiguities that are sporadically tied to the missing period. An illustrative example is shown in *Appendix figure 2*.

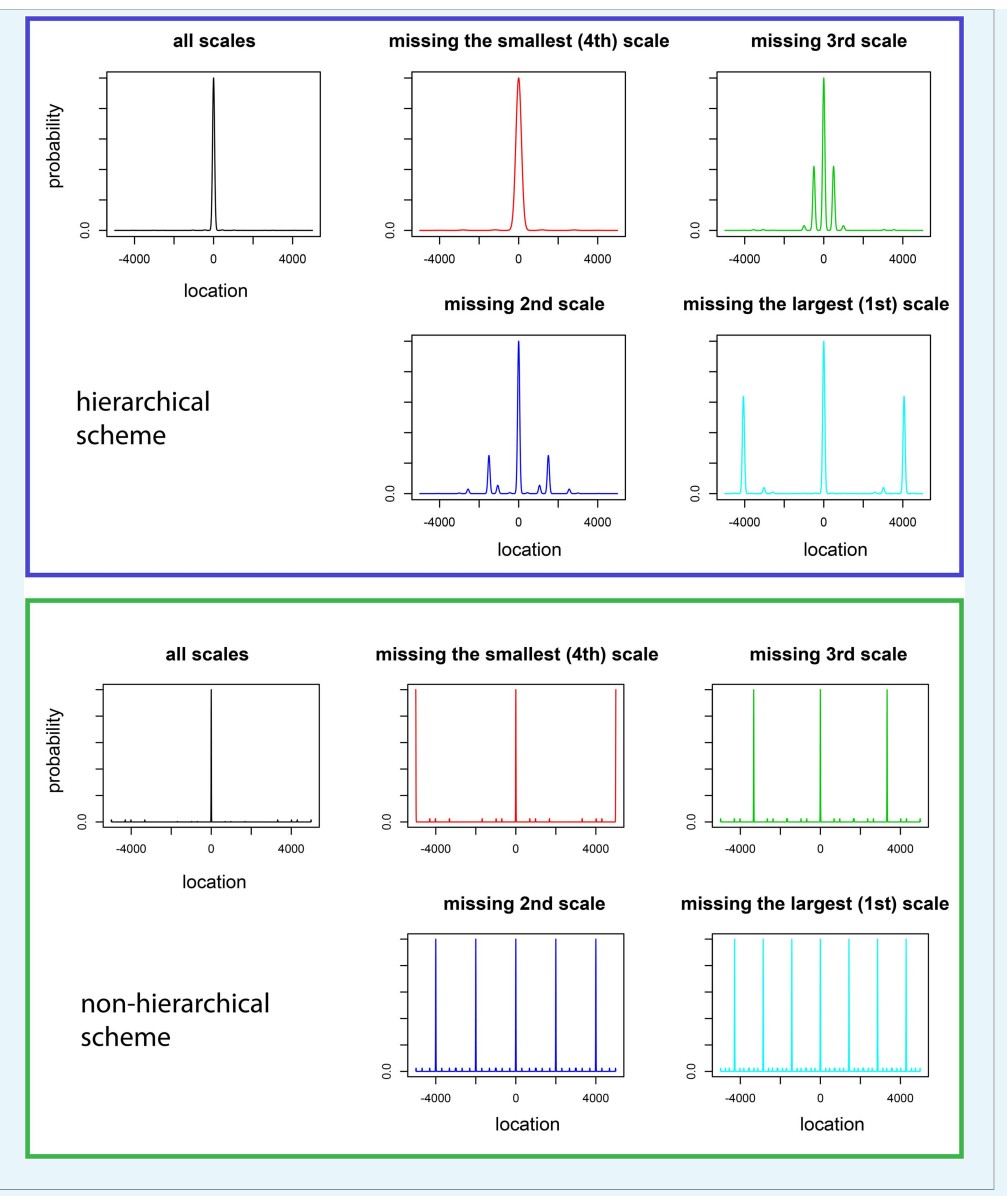

**Appendix figure 2**. The effect of lesioning grid modules on the distribution over location for hierarchical vs non-hierarchical grid schemes. For the hierarchical scheme, we assume that four one-dimensional grid modules are related by a scale factor $r$ ($r = 2.7$), that is, $\frac{\lambda_i}{\lambda_{i+1}} = 2.7$, $i = 1, 2, 3$, and the ratio $\frac{\lambda_i}{\sigma_i} = 9.1$, i = 1, 2, 3, 4. We assume that the animal is at $x = 0$ and construct the probability distribution over location given the activity in each grid module as described in Optimizing the grid system: probabilistic decoder, 'Materials and methods'. For the non-hierarchical scheme, we again assume four grid modules and set the periods of the four modules to be 1/105 (fourth), 1/70 (third), 1/42 (second), 1/30 (first) of the whole range, respectively. We set the width of the composite likelihood after combining all four modules to be 1/210 of the range [−5000, 5000].

The grid cell representation of space in the entorhinal cortex is related in a complex manner to the hippocampal place cell representation (**Bush et al., 2014**; **Sasaki et al., 2015**). Simplistic models of this transformation assume that grid cells are pooled in the hippocampus and that some form of synaptic plasticity selects inputs with the same spatial phase (**Solstad et al., 2006**). In the context of such a model (which does not reflect many aspects of the known physiology), our grid scheme makes specific predictions for the effects of module lesions on place fields.

We use a firing rate model for both place cells and grid cells. The 1-d grid cell firing rate is modeled as a periodic sum of truncated Gaussians (a full Gaussian mixture model gives similar results but the truncated model is easier to handle numerically). We will consider four grid modules with module periods $\lambda_i$, Gaussian standard deviations $\sigma_i$ of the bump of the grid cell tuning curve, and ratios $\lambda_i/\sigma_i = 9.1$. The grid periods follow a scaling $\lambda_i/\lambda_{i+1} = 2.7$, and we examine place coding over the range set by the biggest period $\lambda_1$.

The place cell response is modeled via linear pooling of grid cells with the same phase followed by a threshold and an exponential nonlinearity:

$$f(x) \propto exp\left( \sum_1^4 g_i(x) \right) - c*m.$$

Here, $g_i(x)$ is the grid cell firing rate, $c = 0.3$ sets the threshold and $m = max \{exp(\sum_1^4 g_i(x))\}$ is the maximum activation. This is a simplified description of the essential features of many models of the grid-place transformation (see, e.g., [**Solstad et al., 2006**; **de Almeida et al., 2009**] and the review [**Giocomo et al., 2011b**]). To model the effect of lesioning grid module $i$, we set the $g_i(x) = 0$. The results are shown in the **Appendix figure 3**. Qualitatively, lesioning the smallest grid module increases the place cell width, while lesioning the largest grid module leads to increased firing in locations outside the main place fields. In general, lesioning different grid modules along the hierarchy leads to different effects on the place field. This is a testable prediction in future experiments. Note that lesions of dorsal-ventral bands are not a direct test—multiple grid modules co-exist in each location along the dorsal-ventral axis (**Stensola et al., 2012**).

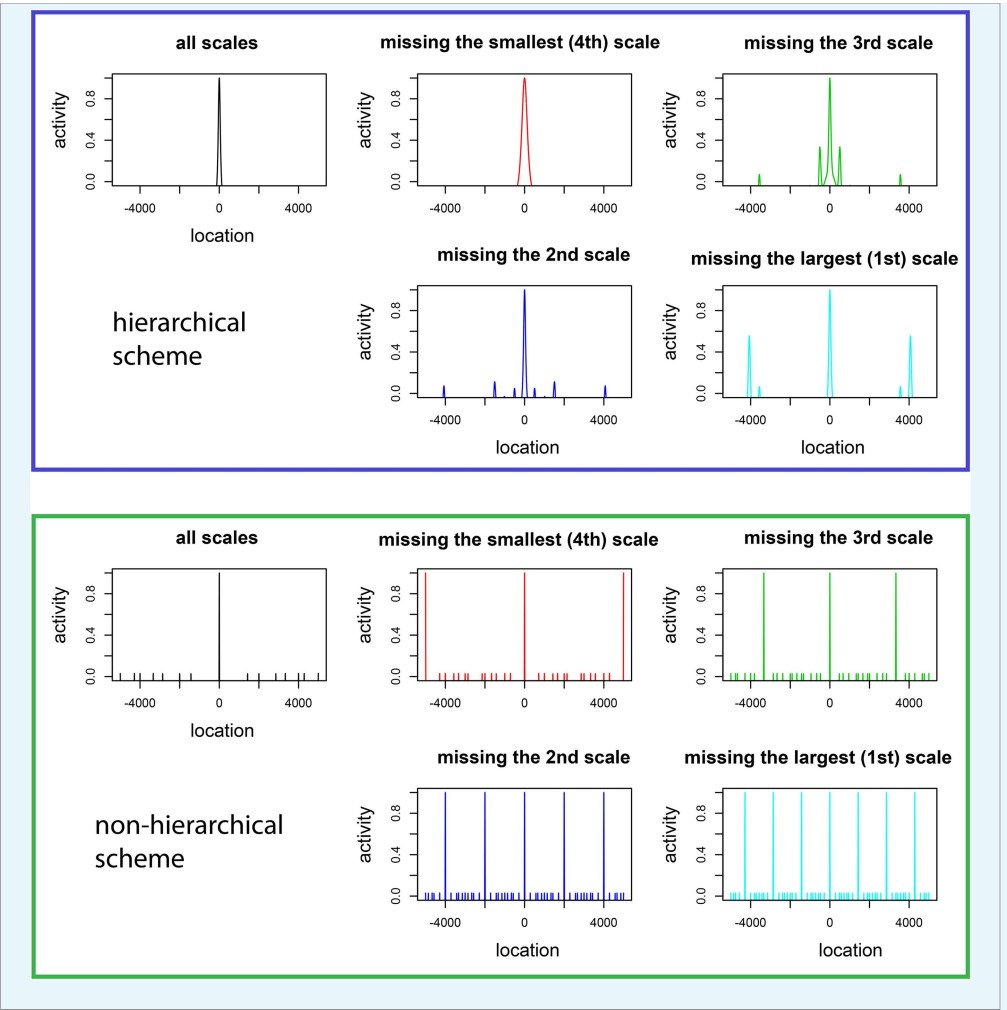

**Appendix figure 3**. The effect of lesioning individual grid modules on place cell activity in a simple grid-place transformation model. Lesioning different modules leads to qualitatively different effects on the place cell response in the hierarchical coding scheme we proposed, as compared to a non-hierarchical scheme. See 'Predictions for the effects of lesions and for place cell activity', Appendix 1 for details.

For comparison purposes, we also simulated a non-hierarchical model where grid periods are similar but incommensurate. In this model, the place cell response is

$$\tilde{f}(x) \propto exp\left( \sum_1^4 \widetilde{g}_i(x) \right) - \tilde{c} * \tilde{m},$$

where $c = \tilde{0}.35$ is a threshold, $\tilde{m} = max \{exp(\sum_1^4 \widetilde{g}_i(x))\}$, and $\widetilde{g}_i(x)$ is the grid cell firing rate again modeled as a sum of truncated Gaussians. In each module, we took the standard deviation of the Gaussians to be 1/210 of the whole range. The periods of the grids in the four modules were 1/105 (forth), 1/70 (third), 1/42 (second), 1/30 (first) of the whole range respectively. Again, to model the effect of lesioning grid module $i$, we set the $\widetilde{g}_i(x) = 0$. In this grid scheme, lesioning any grid module leads to qualitatively similar effects on the place cell activity, as they all lead to the emergence of several place fields (**Appendix figure 3**). This is in contrast with the hierarchical scheme, in which lesioning the largest scale leads to an expansion of place fields rather than an increase in the number of fields.

