## [Decision Letter]

Thank you for submitting your work entitled “A principle of economy predicts the functional architecture of grid cells” for peer review at *eLife*. Your submission has been favorably evaluated by Timothy Behrens (Senior Editor), a Reviewing Editor, and three reviewers.

The reviewers have discussed the reviews with one another and the Reviewing Editor has drafted this decision to help you prepare a revised submission.

In this study, the authors present a simpler, more elegant, and more intuitive presentation (relative to other studies) of coding optimality principles/constraints leading to grid cell formation. While there was agreement about this, two main aspects came forth that should be addressed in a revised submission:

1) Be clear and specific about how this work differs from other studies (e.g. Mathis, Herz et al.). That is, the analysis in the current paper is simpler and does not rely on Fisher information, but rather on quite straightforward assumptions related to the nature of optimal coding.

2) Be clear about how the novel analyses presented (as different from other studies) allow a closer link with experimental studies.

More details of revisions are given below.

Reviewer #1:

The authors present a theoretical analysis of the ideal properties that a “grid coding” system for spatial location should exhibit, by assessing the conditions which allow location to be encoded with maximum precision across a specific spatial range using the minimum number of grid cells. Unlike several previous theoretical studies (most notably, [21]), the authors assume that this spatial range is equal to the scale of the largest grid module (i.e. ∼10m), rather than exploiting the combinatorial properties of the grid cell code to encode location over a range equal to the lowest common multiple of all grid scales. Thus, although a similar topic has been addressed in various guises by several previous publications (i.e. [21]; Mathis et al., 2012, 2013; [58]), there is some novelty to the current work. The manuscript is well written, thorough, and makes some interesting and specific predictions (such as the optimal ratio between grid field size and grid module scale) that have not – to the best of my knowledge – been described elsewhere. Concerns to be addressed are described below.

Specific comments:

In the subsection “Intuitions from a simplified model”, the current body of experimental data in this field simply does not support the authors’ repeated assertion that “[…] anatomical and functional evidence suggests that place cells selectively read out contiguous subsets of entorhinal grid modules along the dorsoventral axis (Van Strien, Cappaert and Witter, 2009; Solstad, Moser and Einevoll, 2006)”. First, citations Van Strien, Cappaert and Witter, 2009 and Solstad, Moser and Einevoll, 2006 are an anatomical review and a theoretical paper, respectively – neither of which can reasonably be described as providing “functional evidence”. Second, several groups have recently published review papers summarising a wide body of functional evidence that directly contradicts the hypothesis that place cells ‘selectively read-out’ a subset of grid cell inputs (i.e. [11]; [46]). The authors should edit the text here, and at several other junctures throughout the manuscript (listed below) to address this point. The hypothesis that place cells represent a read-out of the grid cell system has no bearing on the theoretical work presented, and only serves to misrepresent our current understanding of the grid cell system.

In the Discussion, the authors state that: “Together with the anatomy (Van Strien, Cappaert and Witter, 2009), the hierarchical view of location coding that we have proposed then predicts that dorsal place cells should be revealed to have multiple place fields in large environments because their spatial ambiguities will not be fully resolved at larger scales. Preliminary evidence for this prediction has appeared in [20]; Rich, Liaw and Lee, 2014.” However, those studies show no systematic relationship between the locations of dorsal place cell's multiple firing fields in large environments, which directly contradicts the predictions of a grid cell to place cell model (see, for example, Figure 8 and Appendix–figure 4 in this manuscript). The authors should note this caveat– along with the other experimental data showing that a grid to place cell model is overly simplistic (see above) – or remove the corresponding piece of text.

In the Discussion, the authors “[…] predict that lesioning the modules with small periods will expand place field widths, while lesioning modules with large periods will lead to increased firing at locations outside the main place field, at scales set by the missing module. Our prediction is supported by a recent study demonstrating effects of lesions including dorsal mEC on place field widths in small environments (28)”. This is misleading for several reasons. First, as described above, this statement neglects to mention the wider body of evidence contradicting the hypothesis that grid cell inputs solely generate place cell firing fields. Second, the aim of the cited study (28) was to examine the effect of eliminating all grid cell inputs to place cells, and ∼85% of the total mEC volume was ablated, including 94.6% of layer II and 83.5% of layer III. Are the authors suggesting that the observed effects on place cell firing are a result of remaining grid cell inputs from modules with a large or small period? Third, the more specific prediction is that lesioning grid modules with large periods will lead to the appearance of additional place fields at periodic, grid-like locations in two dimensions, but that analysis is not made or discussed in [28]. Fourth, more recent experimental evidence indicates that focal inactivations of dorsal or ventral mEC each produce place field expansion, and neither generated increased firing at locations outside the main place field – in fact, that data showed a trend towards a decrease in the number of firing fields exhibited by each place cell following focal inactivations (Figure S7 in Ormond et al., 2015). Each of those results also contradict the predictions of a grid cell to place cell model, despite the strange interpretation of the data made in that paper. Hence, the authors should include a citation to that paper and edit the text accordingly.

In the Appendix, the authors again suggest that “the grid cell representation of space in the entorhinal cortex is […] transformed in the hippocampus into the place cell representation.” They should edit this text to more accurately represent the current understanding of the relationship between grid and place cell firing patterns.

Abstract and Introduction: Given that the principal difference between the analysis in this manuscript and that presented in several previous publications (i.e. [21]; [58]) is that the authors assume a spatial range equal to the size of the largest grid module, this should be more explicitly stated in the Abstract and Introduction. This would make the novelty of this manuscript more apparent. For example, the Abstract should be edited to read: “We propose that the grid system implements a hierarchical code for spatial location that economizes the number of neurons required to encode location with a given resolution across a spatial range equal in size to the period of the largest grid module” or similar. Likewise, the Introduction should be edited to read: “minimize the number of neurons required to achieve the behaviourally necessary spatial resolution across a spatial range equal in size to the period of the largest grid module” or similar.

In the Introduction, it is not clear to me what the authors mean by the following statement: “Consistent with studies of grid cell and place cell remapping, our analyses assume that there is a behaviorally defined maximum range over which a fixed grid represents locations (24).” I fail to see the relevance of remapping to the behavioural range of an animal. Could the authors explain their rationale here please?

In the Introduction, when the authors state “three dimensional grids that will be relevant to navigation in, e.g., bats”, they should include a reference to [61], which demonstrates that bats do have grid cell responses, even though they have only been recorded in two dimensional environments so far.

In the subsection “Intuitions from a simplified model”, when the authors stress that “the animal could achieve the required resolution in a place coding scheme […]”, they must incorporate a reference to [21], which makes a very similar comparison of “place coding” and “grid coding” schemes.

Figure 2 is cited before Figure 2 in the main text, which is confusing (subsection “Winner-Take All Decoder”). Similarly, in subsection “General grid coding in two dimensions”, Figure 3 is not referred to in the text at all, and Figure 3 are cited before Figure 2, which is not ideal. It would be preferable if the authors placed all figures pertaining to the 2D case in Figure 3 (i.e. move Figure 2 into Figure 3) and moved Figure 2 to match the flow of the text (i.e. before Figure 2).

In the Discussion, the authors state: “Given homogeneous positive noise correlations within a grid module, which will arise naturally if grid cells are formed by an attractor mechanism, the required number of neurons could be an order of magnitude higher ([50]; Averbeck, Latham and Pouget, 2006)”. It has recently been demonstrated that positive noise correlations appear to be largely absent in the rodent grid cell system, and the authors may wish to note this point and cite the corresponding paper (38) here.

In the Discussion, “the answer depends on the dimension of the grid” should be “the answer depends on the dimensionality of the grid”.

Again, in the Discussion, the authors mention that they “have checked that the optimal grid scheme predicted by our theory, if decoded in the fashion of (21), can represent space over ranges longer than the largest scale”, but do not mention (in the main text) whether it could or not. They should incorporate a brief description of the outcome of those simulations in this section of text, for clarity.

Reviewer #2:

The paper by Wei et al. uses an optimality argument to explain the empirically observed geometric progression of the spacing of grid modules and the ratio of this geometric progression. The paper is nicely written and the arguments are clear. I am however not fully convinced about the potential influence of this paper based on the following reasons.

1) The major assumption of this paper is the ambiguity of the grid cell firing, that is, from the firing of one grid cell, it is not possible to infer in which of the many vertices of the grid one is located. This assumption, however, does not take into account the fact that the peak firing rate of a grid cell at its fields significantly vary. In other words, the translational symmetry is about the positions at which peak firing occurs, not that each field is identical to the other in terms of firing rate. In my view, this experimental fact fundamentally affects the argument offered here.

2) The minimum of the cost function versus ratio of the spacing of successive modules is very wide, raising the question whether one can really say anything meaningful about the value that that ratio should take. It should not escape our attention that in Figure 2, the authors plot the cost function versus the logarithm of the ratio between successive modules, which gives the impression of a narrower minimum (though still wide). Even with this, the authors’ prediction is stated to “[…] robustly lie in the rage 1.4-1.7[…]”. This is a 20% range.

3) (a) The idea of using optimality for predicting the ratio of grid spacing of the modules has been already employed by Mathis et al., Neural Comp 2012. There are differences between the two works, e.g. Mathis et al. maximize the resolution given a fixed number of neurons while Wei et al. minimize the number of neurons given the resolution and Mathis et al. only focus on the one dimensional case. Despite the differences, it is not clear what is the major conceptual advancement. As far as I can say, the argument of Mathis et al. can be easily extended to 2D to produce a geometric progression.

(b) It is true that, as stated in the in the conclusion, in the work of Mathis et al. the optimal ratio depends on “the number of neurons per module and peak firing rate”. But the prediction of the optimal ratio here also varies over a wide range, depending on the assumption on the decoding scheme (and probably the shape of the tuning curves, assumption on the correlation between neurons etc.).

Reviewer #3:

This excellent paper uses a very simple principle for demonstrating that coding of grid cells is better than coding of place cells, and generates some postdictions following this simple principle. The basic idea is that grid cells act as a kind of “Base-b” representation of space, and it is shown that the representation is optimal when the base chosen is base e (2.71828…). From that, various postdictions follow (which conform nicely with known experiments). Specifically, grid cell modules have a constant scale ratio, which should be √*e* in the simplest model, and closer to the real experimental value (1.4) in a probabilistic model of the cells coding. Furthermore, there should be a certain optimal ratio between the grid field width and the spacing between grid points.

The paper interacts nicely the papers of the group of Andreas Herz, which deal with similar issues using Fisher information. I have no major concerns, as I think the paper is well written, deals with an important subject, looks sound mathematically, and has a nice treatment of relation to experimental data.

The only issue I would like to be dealt with is to make the Discussion more clear as to the relation between this paper and the papers from the Herz group (including the relevant recent one from 2015). Specifically, they have a treatment of the issue of grid cell coding through Fisher information, and it could be of value to connect the work performed here to their line of thought, at least minimally by adding some discussion to the paper (elaborating on the existing paragraph).

Another small question I am curious about is whether the winner-take-all decoder could be seen as a limit-case of the probabilistic decoder. But if that is the case, I do not completely understand the “leap” from *e* to 2.4.

---

## [Author Response]

*In this study, the authors present a simpler, more elegant, and more intuitive presentation (relative to other studies) of coding optimality principles/constraints leading to grid cell formation. While there was agreement about this, two main aspects came forth that should be addressed in a revised submission*.

*1) Be clear and specific about how this work differs from other studies (e.g., Mathis, Herz et al.). That is, the analysis in the current paper is simpler and does not rely on Fisher information, but rather on quite straightforward assumptions related to the nature of optimal coding*.

To respond to this recommendation, we have expanded our discussion of what sets our work apart from others, especially Mathis et al. The comments of the referees have been helpful in this regard. We have attempted to be clear and specific that Mathis et al. explored grid coding in one dimension using Fisher information and numerical simulation to explore decoding error. They found that the set of periods that maximizes the Fisher information is approximated by a geometric series in a regime of large-scale ratios. By contrast, we rely on a simpler formulation of optimal coding with straightforward assumptions about tradeoffs between ambiguity and resolution in a hierarchical grid. We take a simpler definition of the resolution (as the largest scale divided by the smallest scale the system can discriminate), assume that the grid encodes location with a restricted range, and then seek to minimize the resources (number of neurons) required to achieve a given resolution within this range. Our simpler formulation allows us to extend our analysis to any number of dimensions, and to predict the values of structural parameters of the grid such as the ratio between periods, the grid geometry etc. We have added substantially to the Discussion to address these points.

In more detail, the Fisher information approximation to position error in Mathis et al. is only valid over a certain range of parameters. They introduce a no-ambiguity constraint to keep them within this range, but this creates two challenges for an optimization procedure: (1) The optimum depends on the details of the constraint, which was somewhat arbitrarily chosen and dependent on the variability, the tuning curve shape of grid cells and a “tolerable error level”, and (2) The optimum turns out to saturate the constraint, so for some choices of the constraint the procedure is pushed right to the edge of where the Fisher information is a valid approximation at all, causing difficulties for the self-consistency of the procedure. Because of these limits on the Fisher information approximation, Mathis et al. proceed to measure decoding error directly through their numerical studies. But here a complete optimization is not possible because there are too many inter-related parameters. This last point is a limitation of any numerical study. In contrast, we estimated decoding error directly by working with approximated forms of the posteriors rather than by approximating decoding error in terms of the Fisher information. For the winner-take-all analysis, we effectively approximate posteriors as periodic boxcar functions, for the probabilistic analysis as periodic sums-of-Gaussians. These choices allow analytical treatment of the optimization problem over a much wider parameter range without requiring arbitrary hand-imposed constraints.

While going through the paper of Mathis and collaborators carefully for the purpose of this revision, we have also developed some concerns about their analysis. First, with the assumptions as formulated in their paper, and the scores of neurons that are known to exist in each grid module, the scale ratios would generically be predicted to be much larger than 1 (please see the detailed response to Reviewer 2). This is in tension with data. Even ignoring this point, optimizing the Fisher information generally predicts a hierarchy of scale ratios, and only predicts geometric scaling if that scale is significantly greater than 1. Experimentally the scale ratio is ∼1.5. Thus, it seems that optimizing Fisher information does not predict geometric scaling in the regime of relevance to experiment. What is more, while Mathis et al. can only predict a geometric scaling if the scale is large, their Figure 5 illustrates that for large scale ratios the Fisher information does a poor job in approximating the decoding error. So this means that their prediction of a geometric series of periods, even in one dimension, has a limited range of validity. The optimal one-dimensional grid in our work is perched near the edge of the estimated range where their analysis appears to be valid.

We have described these last points in the detailed response to the referees. However, pointing out limitations of Mathis et al. is not our goal in this paper. Hence, we have simply added the phrase “in a regime where the scale factor is sufficiently large” in our description of their prediction of geometric scaling in one dimension, without further comment on the limitation of this prediction.

*2) Be clear about how the novel analyses presented (as different from other studies) allow a closer link with experimental studies*.

We have expanded our discussion of the link with experiments. Specifically, we have pointed out that, as distinct from other studies, our approach allows us to predict that: (a) grid fields should lie on a triangular latice, (b) grid periods should follow a geometric projection, (c) the ratio between grid scales should be *e*^*1/2*^ for idealized neurons, liying between 1.4 and 1.7 for realistic neurons, (d) the scale ratio should vary modestly within and between animals, (e) the optimal scale ratio in one and three dimensions. With some additional assumptions we also predict: (i) the number of grid modules should be ∼10, and (ii) the ratio between grid periods and field widths. Finally, we examine possible deficits in spatial behavior that will obtain upon inactivating grid modules in the context of specific models of grid cells readout. Most of this material is in the Abstract, Introduction, Comparison to Experiment and Discussion sections.

In the original submission, we used a simple model of linear summation of grid cells to make place cells to investigate the effects of grid module inactivation in a hierarchical grid system like the one we study. Reviewer 1 pointed out that: (a) the idea that place cells are a read-out of grid cells has no direct bearing on our theoretical work, (b) recent experimental work very strongly suggests that while the hippocampal place system is certainly affected by the grid system, place cells are not a “read-out” of the grid system in any simple sense of that term. We agree entirely and have edited the text in detail to reflect these points. The changes are in Results, Discussion and Appendix.

Details of the changes are described below.

Reviewer #1:

*[…] In the subsection “Intuitions from a simplified model”, the current body of experimental data in this field simply does not support the authors’ repeated assertion that “[…] anatomical and functional evidence suggests that place cells selectively read out contiguous subsets of entorhinal grid modules along the dorsoventral axis (Van Strien, Cappaert and Witter, 2009; Solstad, Moser and Einevoll, 2006)”. First, citations Van Strien, Cappaert and Witter, 2009 and Solstad, Moser and Einevoll, 2006 are an anatomical review and a theoretical paper, respectively – neither of which can reasonably be described as providing “functional evidence”. Second, several groups have recently published review papers summarising a wide body of functional evidence that directly contradicts the hypothesis that place cells ‘selectively read-out’ a subset of grid cell inputs (i.e.*
[11]*;*
[46]*). The authors should edit the text here, and at several other junctures throughout the manuscript (listed below) to address this point. The hypothesis that place cells represent a read-out of the grid cell system has no bearing on the theoretical work presented, and only serves to misrepresent our current understanding of the grid cell system*.

We agree that, as written, our paper suggests an understanding of the relation between grid and place cells that is both overly definitive, and one which is challenged by recent findings (e.g. that place cells are active before grid cells, and that place fields survive sustained inactivation of grid cells). The material that the referee is commenting on arose from multiple discussions with audiences of seminars and readers of our manuscript. We were repeatedly asked (and are still asked during talks) how our view of a hierarchical grid code would affect readout of the grid system, perhaps via place cells, as compared to a non-hierarchical grid system of the sort proposed by Burak and Fiete. We decided that a concrete way of addressing these questions would be to pick a simple model relating grid and place cells (e.g. the Fourier summation setup of Solstad et al., and explicitly show that different grid schemes can have different effects).

However, we fully agree with the reviewer that: (a) the idea that place cells are a read-out of grid cells has no direct bearing on our theoretical work, (b) recent experimental work very strongly suggests that while the hippocampal place system is certainly affected by the grid system, place cells are not a “read-out” of the grid system in any simple sense of that term. The two reviews cited above ([11] and [46]) make the latter point very clearly and effectively.

As we see it there are two options: (1) we could completely remove any mention of a readout via place cells or otherwise and simply discuss the architecture of the grid system; (2) we could be clear that we are going to look at linear summation models of grid cell readout as a toy model, not because we think they accurately represent the readout, but to show that different assumptions about the grid architecture can lead to different specific effects for manipulations like lesions. Of course, in order to make specific predictions for how lesions in the grid system would affect place cells, we would need detailed knowledge about the precise relationship between these systems, and while there are many hints of a complex relationship, the details remain unclear.

We decided to go with option (2), because we are repeatedly asked, “I know that the relation between place cells and grid cells is complicated, but can you tell me what would happen if you imagine, for purposes of argument, a simple linear summation readout and then remove some modules?” So we think that including this material (which is mostly in the Appendix and Discussion), with appropriate nuance and caveats, might be helpful to some readers. We have revised to try to achieve this goal, but are open to the idea of simply leaving this material out.

We have made a series of changes starting with the remark that the referee mentions (in the subsection “Intuitions from a simplified model”) and continuing throughout the paper (please also see our responses to the comments below). Also, a minor point – as the referee says, references to Van Strien, Cappaert and Witter, 2009, and Solstad, Moser and Einevoll, 2006 are an anatomical review and a theory paper, and we have modified the citation accordingly.

*In the Discussion, the authors state that: “Together with the anatomy (Van Strien, Cappaert and Witter, 2009), the hierarchical view of location coding that we have proposed then predicts that dorsal place cells should be revealed to have multiple place fields in large environments because their spatial ambiguities will not be fully resolved at larger scales. Preliminary evidence for this prediction has appeared in*
[20]*; Rich, Liaw and Lee, 2014.” However, those studies show no systematic relationship between the locations of dorsal place cell's multiple firing fields in large environments, which directly contradicts the predictions of a grid cell to place cell model (see, for example,*
Figure 8
*and Appendix–figure 4 in this manuscript). The authors should note this caveat– along with the other experimental data showing that a grid to place cell model is overly simplistic (see above) – or remove the corresponding piece of text*.

The studies of [20], and [44], find that dorsal place cells often have multiple place fields. Fenton et al. claim that 85% of dorsal CA1 place cells have multiple place fields, and the majority of cells in Rich et al. have multiple place fields in a 48m track, some having dozens of place fields. However, these studies show that the locations of these place fields are sporadic, perhaps even random according to some distribution. This disordered structure may be in tension with a simplistic summation view of the grid-to-place cell transformation. On the other hand, it should be noted that the data of Stensola et al. shows that there is significant variability of the period, orientation and ellipticity within each module. As part of a different collaboration, one of us (VB) has been investigating the consequences of this variability for spatial coverage – it seems to produce significant differences in the relative phase of grid cells between unit cells of the grid lattice. This variability can change the prediction of regularity in the locations of multiple place fields in a naive summation model. That said, investigating this properly lies outside the scope of the present paper. Thus we have contented ourselves with adding appropriate nuance and caveats as follows: (a) indicate that a naive summation model of place cells along with the anatomy of mEC-hippocampal projections predicts multiple place fields for a single dorsal place cell, as seen in experiments, (b) a naive model of this kind also predicts a orderly distribution of place fields which is not seen, (c) however, the variability in the grids even within a module likely interferes with the predicted order, (d) and in any case there is significant evidence (see Bush et al., and Sasaki et al. for a summary) that place cells are not formed and maintained by grid cells alone. The changes have been added to the sixth paragraph of the Discussion.

*In the Discussion, the authors “[…] predict that lesioning the modules with small periods will expand place field widths, while lesioning modules with large periods will lead to increased firing at locations outside the main place field, at scales set by the missing module. Our prediction is supported by a recent study demonstrating effects of lesions including dorsal mEC on place field widths in small environments (*[28]*)”. This is misleading for several reasons. First, as described above, this statement neglects to mention the wider body of evidence contradicting the hypothesis that grid cell inputs solely generate place cell firing fields. Second, the aim of the cited study (*[28]*) was to examine the effect of eliminating all grid cell inputs to place cells, and ∼85% of the total mEC volume was ablated, including 94.6% of layer II and 83.5% of layer III. Are the authors suggesting that the observed effects on place cell firing are a result of remaining grid cell inputs from modules with a large or small period? Third, the more specific prediction is that lesioning grid modules with large periods will lead to the appearance of additional place fields at periodic, grid-like locations in two dimensions, but that analysis is not made or discussed in*
[28]*. Fourth, more recent experimental evidence indicates that focal inactivations of dorsal or ventral mEC each produce place field expansion, and neither generated increased firing at locations outside the main place field – in fact, that data showed a trend towards a decrease in the number of firing fields exhibited by each place cell following focal inactivations (Figure S7 in Ormond et al., 2015). Each of those results also contradict the predictions of a grid cell to place cell model, despite the strange interpretation of the data made in that paper. Hence, the authors should include a citation to that paper and edit the text accordingly*.

We agree entirely that there is a substantial body of evidence that grid cells do not solely generate place cells, although they do influence some of the functional properties of the hippocampus. Further, Hales et al. were indeed attempting to examine the effects of eliminating all the grid inputs and found that the substantial ablation they performed led to fewer, smaller and less stable place fields. These ablations were not specific to a given module, but we would expect in a hierarchical grid code that elimination of many contributions with small periods would decrease the precision of spatial coding that exploits grid cell responses. Of course elimination of large periods should lead to ambiguities in large environments and Hales et al. did not test for or discuss this – they are working with small 1m x 1m environments so we might not expect to see many ambiguities.

There is also the paper of Ormond et al., which discussed focal inactivations. We find the results in this paper difficult to interpret also. For starters, the inactivations are focal along the dorso-ventral axis of the mEC, but the data in Stensola et al. seem to indicate that that many grid modules are present at every mEC depth, with a dorsal enrichment of small periods. So the focal inactivations of Ormond et al. would seem to still be inactivating modules with multiple periods. Thus, even in a naive summation model, we would expect expansion of place fields for all of these inactivations with larger expansion for dorsal fields. This is because dorsal lesions would get rid of more cells in the smallest modules, and would lead to a greater broadening. Ventral lesions would still remove some cells with small periods, and so the broadening effect should be smaller as Ormond et al. appear to see.

The data in the supplement of Ormond et al. seem to indicate, on the one hand, a decrease in the number of distinct place fields associated to place cells (in tension with a grid to place model), but on the other hand they seem to show an increase in the “out-of-field” firing rate. What is more, these figures likely include various cases where the place cells went from having one field to having none (i.e. the place field simply disappeared) – it is hard to be sure, because the information was not provided as far as we can tell, and thus it is hard to evaluate whether this is in fact inconsistent with a summation model. One could also imagine that the general increase in the size and noisiness of place fields might lead to a decrease in the number of place fields that can be accommodated in the 7m linear track that they were working with. In particular, the decrease in theta power makes their measurements much more prone to noise. Finally, Ormond et al. are recording dorsally in CA1 and CA3. If the anatomy of Witter et al. that we cite is accurately indicative of functional connectivity, then ventral inactivation in mEC would have less clear effects on these dorsal hippocampal recordings. That makes it still more problematic to interpret what is going on. We also note that none of the individual examples depicted in Figure S5 of that paper show a decrease in ambiguity after lesions. So we conclude from Ormond et al. that there is no clear evidence for a decrease in ambiguity following lesion, but it is similarly debatable whether there is any indication of an increase in ambiguity.

Given the complex, and, in our view, difficult to interpret, experimental situation we have edited as follows. First, we have modified our remarks to make it clear that we regard a linear summation model of place cells as simplistic in view of recent experimental developments reviewed in Bush et al. and Sasaki et al. Second, we are more clear that we are simply seeking to illustrate that different grid schemes can have different effects on specific readouts. Finally we refer to Hales et al. and Ormond et al. in a nuanced way, clarifying that these experiments do not lesion individual modules and thus do not constitute specific evidence for the results of such lesions. These changes have been added to the Discussion section.

*In the Appendix, the authors again suggest that “the grid cell representation of space in the entorhinal cortex is […] transformed in the hippocampus into the place cell representation.” They should edit this text to more accurately represent the current understanding of the relationship between grid and place cell firing patterns*.

We have edited this text to be more nuanced and accurate about the current state of understanding of the relationship between the mEC and the hippocampus (see Section F of the Appendix). Specifically we say: “The grid cell representation of space in the entorhinal cortex is related in a complex manner to the hippocampal place cell representation (11, 46). […] In the context of such a model (which does not reflect many aspects of the known physiology), our grid scheme makes specific predictions for the effects of module lesions on place fields.”

The papers mentioned above are now cited in the main text, and again in the Appendix.

*Abstract and Introduction: Given that the principal difference between the analysis in this manuscript and that presented in several previous publications (i.e.*
[21]*;*
[58]*) is that the authors assume a spatial range equal to the size of the largest grid module, this should be more explicitly stated in the Abstract and Introduction. This would make the novelty of this manuscript more apparent. For example, the Abstract should be edited to read: “We propose that the grid system implements a hierarchical code for spatial location that economizes the number of neurons required to encode location with a given resolution across a spatial range equal in size to the period of the largest grid module” or similar. Likewise, the Introduction should be edited to read: “minimize the number of neurons required to achieve the behaviourally necessary spatial resolution across a spatial range equal in size to the period of the largest grid module” or similar*.

Thank you for this suggestion. We have made this change, and it helps to clarify the differences between the frameworks.

We have also added a reference to Towse et al. (please see the Discussion section) as part of our analysis of the relation of our study to previous work: “Note that decoding error was also studied in Towse et al. and those authors reported that the results did not depend strongly on the precise organization of scales across modules.”

*In the Introduction, it is not clear to me what the authors mean by the following statement: “Consistent with studies of grid cell and place cell remapping, our analyses assume that there is a behaviorally defined maximum range over which a fixed grid represents locations (*[24]*).” I fail to see the relevance of remapping to the behavioural range of an animal. Could the authors explain their rationale here please?*

We intended to say that assuming that the grid code can only represent location up to some maximum range without additional information, it would be necessary that a new grid should be loaded upon reaching the edge of the representational range. The ability of grids to remap in new environments suggests that this should be possible. In effect, we were trying to suggest that an animal could “stitch together” a representation of a very large environment by remapping its grids between segments. To keep things simple we have removed the phrase “Consistent with studies of grid cell and place cell remapping”.

*In the Introduction, when the authors state “three dimensional grids that will be relevant to navigation in, e.g., bats”, they should include a reference to*
[61]*, which demonstrates that bats do have grid cell responses, even though they have only been recorded in two dimensional environments so far*.

We have added the citation.

*In the subsection “Intuitions from a simplified model”, when the authors stress that “the animal could achieve the required resolution in a place coding scheme […]”, they must incorporate a reference to*
[21]*, which makes a very similar comparison of “place coding” and “grid coding” schemes*.

We have added this citation. Thank you for the suggestion.

Figure 2
*is cited before*
Figure 2
*in the main text, which is confusing (subsection “Winner-Take All Decoder”). Similarly, in subsection “General grid coding in two dimensions”,*
Figure 3
*is not referred to in the text at all, and*
Figure 3
*are cited before*
Figure 2*, which is not ideal. It would be preferable if the authors placed all figures pertaining to the 2D case in*
Figure 3
*(i.e. move*
Figure 2
*into*
Figure 3*) and moved*
Figure 2
*to match the flow of the text (i.e. before*
Figure 2*)*.

Thank you for these suggestions. In order to respect the flow of the text we have reorganized as follows. We moved Figure 2 (the optimization curve in 1D) to be a panel of Figure 6 where other material on the optimization in one dimension is gathered. We removed Figure 3 (which was not referred to) and we moved Figure 2 into Figure 3. Now Figure 2 is focused on illustrating the precision-ambiguity tradeoff in the setting of probabilistic decoding. Figure 3 is focused on the two dimensional optimization. We hope that this helps with clarity.

*In the Discussion, the authors state: “Given homogeneous positive noise correlations within a grid module, which will arise naturally if grid cells are formed by an attractor mechanism, the required number of neurons could be an order of magnitude higher (*[50]*; Averbeck, Latham and Pouget, 2006)”. It has recently been demonstrated that positive noise correlations appear to be largely absent in the rodent grid cell system, and the authors may wish to note this point and cite the corresponding paper (*[38]*) here*.

Thank you for this suggestion. We have added this citation. However, the authors of the paper seem to explicitly say that they did find noise correlations. More specifically, their abstract says: “We analyze the noise correlations between pairs of grid code neurons in behaving rodents. We find that if the grids of the two neurons align and have the same length scale, the noise correlations between the neurons can reach 0.8. For increasing mismatches between the grids of the two neurons, the noise correlations fall rapidly.” This is apparently also the message that they derive from their Figure 9. Meanwhile Dunn, Morreaunet and Roudi (2015) also report positive noise correlations for grids with similar phases, and vanishing or sometimes negative correlations for grids with very different phases. Since most grid cells differ in their mutual phase or period, we take the referee’s point. Since the presence or absence of noise correlations is not a main point of our paper, we have simply indicated (Discussion, fourth paragraph) that Mathis et al. and Dunn et al. investigated noise correlations between grid cells and found positive correlations for aligned grids (i.e. similar phase) of the same scale and weak correlations otherwise.

*In the Discussion, “the answer depends on the dimension of the grid” should be “the answer depends on the dimensionality of the grid”*.

We have made this change.

*Again, in the Discussion, the authors mention that they “have checked that the optimal grid scheme predicted by our theory, if decoded in the fashion of (*[21]*), can represent space over ranges longer than the largest scale”, but do not mention (in the main text) whether it could or not. They should incorporate a brief description of the outcome of those simulations in this section of text, for clarity*.

We were trying to indicate the following. The optimization analysis predicts a particular scale ratio and enough neurons in each module to ensure that the likelihood function over position in each module has peak widths that are a certain fraction of the period. The range of representation can be extended by shrinking the widths of the likelihood ratio peaks. This requires increasing the number of neurons in each module beyond the minimum required for the spatial range that we started with. We have edited this text to say: “Nevertheless, we have checked that a grid coding scheme with the optimal scale ratio predicted by our theory can represent space over ranges larger than the largest grid period (Appendix, Section E). However, to achieve this larger range, the number of neurons in each module will have to increase relative to the minimum in order to shrink the widths of the peaks in the likelihood function over position.” The edited text is in the Discussion.

Reviewer #2:

*1) The major assumption of this paper is the ambiguity of the grid cell firing, that is, from the firing of one grid cell, it is not possible to infer in which of the many vertices of the grid one is located. This assumption, however, does not take into account the fact that the peak firing rate of a grid cell at its fields significantly vary. In other words, the translational symmetry is about the positions at which peak firing occurs, not that each field is identical to the other in terms of firing rate. In my view, this experimental fact fundamentally affects the argument offered here*.

Consider the probabilistic decoder. In this case, *P(x|i)* can be approximated as a periodic sum of Gaussians without making restrictive assumptions about the shapes of the tuning curves of individual grid cells, or about the precision of their periodicity, so long as, on average, the variability of individual neurons is weakly correlated and homogeneous.

For example, even though individual grid cells can have somewhat different firing rates in each of their firing fields, this spatial heterogeneity will be smoothed in the posterior over the full population of cells, leading to much more accurate periodicity. In other words, individual grid cells show both spiking noise and “noise” due to heterogeneity and imperfect periodicity of the firing rate maps. Both these forms of variability are smoothed out by averaging over the population, provided there are enough cells and noise is homogeneous and not too correlated – we assume this. The first paragraph in the “Probabilistic Decoder” subsection makes these points.

Even if the experimentally-measured heterogeneity is too strong to be completely neglected, we still feel that our framework provides value in studying the grid system. Developing a theoretical framework that solves the simpler case of perfect periodicity is a natural starting point for studying the more complex, realistic case. Experimental details that deviate from our idealized assumptions may be added, and our calculations modified to see how these complications modify our predicted optimality conditions. We think this is an exciting avenue for future work building on the results and framework we have reported here.

*2) The minimum of the cost function versus ratio of the spacing of successive modules is very wide, raising the question whether one can really say anything meaningful about the value that that ratio should take. It should not escape our attention that in*
Figure 2*, the authors plot the cost function versus the logarithm of the ratio between successive modules, which gives the impression of a narrower minimum (though still wide). Even with this, the authors’ prediction is stated to “[…] robustly lie in the rage 1.4-1.7[…]”. This is a 20% range*.

Please note that our text goes to some pains to point out that the minima of the cost functions are not extremely sharp (in the subsection “General grid coding in two dimensions”). In our view this is a virtue, not a problem, because it means that a degree of variability in the grid parameters can be tolerated. Please also note that, as we say, the predictions of the simple winner-take-all and probabilistic models lie within the “overlapping shallow basins” of the two models. Given that these two models lie at extremes of decoding complexity, this adds to our confidence that over a wide range of assumptions the optimal grids will lie within a similar range. Similar considerations apply to both the one dimensional and two dimensional grids.

As we also state, the relative shallowness of the minima lead us to expect that the parameters of the grid should be somewhat variable between between cells within a module, and between individuals. Indeed, the experimental measurements are variable in this way. It is difficult to formulate a theory of precisely how much variability, and associated cost in the number of neurons, is acceptable to the animal. In this situation, the sensible prediction to make is that the grid periods ratios will be localized around a certain value, and to ask what deviation in cost relative to the optimum is implied by the experimentally determined spread of these ratios (we find a ∼5% deviation in cost). This is what we have done in Figure 4.

We can further illustrate these points by considering an additional kind of variability in the experimentally measured grids. It has been noted that grids can have an ellipticity – i.e. they can be “squished”. Our analysis of grid geometries in two dimensions showed that the triangular grid is optimal, but geometries close to the triangular one will do well also (see the contour plot in Figure 3). How does the range of ellipticity in the experiments compare to the tolerable extent predicted by the theory?

To address this point, we can re-examine the contour plot in Figure 3 which shows *N/N*_*min*_ (number of neurons/minimum number of neurons) as a function of the array geometry after minimizing over the scale factors between modules for a fixed resolution *R*. The plateau around the triangular array geometry (the point in the middle of the plot) shows that a range of ellipticities will be similarly efficient. To show this range explicitly in a different way, we can keep *N* fixed, and plot the logarithm of the resolution (as defined in the main text), normalized to its maximum, which is achieved at the optimal triangular grid – we will call this the “relative efficiency”. The array geometry is parametrized in terms of two variables, v parallel and v perpendicular as described in the main text, with the triangular lattice being given by *v∥*= 1/2, *v⊥*= √3/2. The plot below shows the relative efficiency as a function of *v∥* for *v⊥*= 1/2 – there is a plateau surrounding the triangular lattice parameters, with a sharp decline in efficiency on either side. Marked on the figure is a range of ellipticities 1.0-1.4 that is wider than the range reported in [52] (the largest ellipticity there was 1.26, albeit with a small sample). Satisfyingly, the ellipticities reported by Stensola et al. will all be closely arranged along the plateau, as our theory would predict.

Author response image 1.**DOI:**
http://dx.doi.org/10.7554/eLife.08362.013

We have not included this analysis (Author response image 1) in the paper because it is not comprehensive and we hope to include a more detailed version in work that is ongoing. But we hope that the result helps to answer the referee’s question.

*3) (a) The idea of using optimality for predicting the ratio of grid spacing of the modules has been already employed by Mathis et al., Neural Comp 2012. There are differences between the two works, e.g. Mathis et al. maximize the resolution given a fixed number of neurons while Wei et al. minimize the number of neurons given the resolution and Mathis et al. only focus on the one dimensional case. Despite the differences, it is not clear what is the major conceptual advancement. As far as I can say, the argument of Mathis et al. can be easily extended to 2D to produce a geometric progression*.

First, we would like to be clear that, contrary to the assertion here, Mathis et al. *did not claim to predict a value for the ratio of grid spacings in grid modules in any dimensions, and did not attempt to say anything about the optimal period ratio and grid shape in two dimensions.* They formulated the Fisher information for decoding position from populations of periodic, one dimensional tuning curves and found that under some conditions the set of periods that maximizes the Fisher information approximates a geometric series. However, as discussed in detail below, their derivation, as they present it: (1) generically implies either unrealistically large period ratios or unreasonably small numbers of cells in each module, both of which disagree with experiment, and (2) in general does not predict a constant scale ratio *r* unless *r >> 1*, which it is not in the data. Incidentally, the derivations in Mathis et al. also have minor mathematical errors which are fixed in the discussion below.

Mathis et al. write an expression for the Fisher Information of the *i*th module which takes the form (Equation 3.22):

J_i_ = C_1_ (M_i_/λ_i_)^2^ (1)

where C_1_ is a constant, M_i_ is the number of cells in module i, and λ_i_ is the period of module i. The sum of M_i_ over *L* modules is *N*, the number of cells in the grid system. The total Fisher Information J is the sum of the J_i_. In any treatment of the grid system we must understand how information from different modules is integrated to eliminate the ambiguity in position left by the responses in a single module. Mathis et al. resolve this ambiguity by a hard constraint (which is reminiscent of our Winner-Take-All model) by setting (Equation 3.24 and the text below it):

λ_i+1_ = D(ε)/(J_i_)^1/2^ = (D(ε)/M_i_ C_1_) λ_i_ (2)

Here D(ε) is a “safety factor” that depends on noise and the tuning curve shapes. They arrive at this equation by first placing a bound on how small the period of module *i+1* can be to achieve a tolerable degree of ambiguity (set by D(ε)) and then saying that the Fisher Information is optimized when this bound is met.

But Equation (2) above implies that the ratio of scales that we seek to predict has the form:

r_i_ = λ_i_/λ_i+1_ = M_i_ (C_1_/D(ε)) (3)

Now Mathis et al. are taking C_1_/D(ε) to be a parameter of O(1) (bottom of their p. 17). But M_i_, the number of grid cells in each module is expected to be in the scores or maybe the hundreds. So, given the general assumptions of Mathis et al., the period ratio r_i_ would be expected to be much larger than 1, contrary to experiment. Alternatively, to get an O(1) scale ratio with C_1_/ D(ε) ∼ O(1), you could take M_i_ to be O(1). This seems to be the scenario considered in Mathis et al., where they say that M_i_ ∼ 3 would be optimal. But we know that each module contains many more cells than that. A final option may be to suppose that C_1_/D(ε) is small in their formalism. But they do not seem to consider this in their paper. *Hence we conclude that the analysis of Mathis et al., as presented in their paper, gives estimates that are in tension with data.*

Ignoring this for the moment, we can proceed further with their analysis. As far as we can tell there is a minor mathematical error in going from their Equation (3.26) to their Equation (3.27) for the population Fisher Information. They chose to scale M_i_ as M_i_’= (C_1_/ D(ε))^1/2^ M_i_, but this does not lead the scaling in Equation 3.27. The correct choice seems to be to scale M_i_ as P_i_ = M_i_ C_1_/D(ε). This difference leads to different constant factors in front of Equation 3.27 and different scalings of variables in their analysis of regimes of validity. Neither of these changes makes a big difference to the analysis, but it is worth correcting the small error anyway. In any case, the bottom line is that the Fisher Information can be written as:

J = (D(ε)/λ_0_)^2^ (r_0_^2^ + r_0_^2^r_1_^2^ + r_0_^2^r_1_^2^r_2_^2^ +…) (4)

This is (Equation 3.27) in Mathis et al. rewritten in terms of the scale ratios. To get the coefficients in front right you have to fix the minor scaling error mentioned above. Meanwhile, the constraint on the total number of cells is:

N (C_1_/D(ε)) = r_0_+r_1_+r_2_+… (5)

This is simply the constraint on the sum of M_i_ written in terms of Equation (3) above.

It is obvious that (4) is not symmetric between the r_i_ and hence the Fisher Information equations of Mathis et al. *do not in general predict a geometric series of periods (i.e. constant r*_*i*_*)*. In fact, one can show by optimizing (4) above with a Lagrange multiplier imposing the constraint (5) that:

r_i_>r_i+1_ (6)

when the Fisher Information is optimized. (For example, in the case of two scales the problem becomes maximizing r_0_^2^+ r_0_^2^r_1_^2^ subject to the constraint r_0_ + r_1_ = constant. If r_1_ is large we can ignore the contribution from the first term, and optimizing gives r_0_=r_1_.

However, the additional r_0_^2^term favors making r_0_ slightly larger as compared to making r_0_ and r_1_ equal.) So the scale ratios are *not* all equal at the optimum. Of course, we can try to get a symmetric solution optimizing the Fisher Information J in Equation (4) by supposing that the r_i_ are much larger than 1, so that the symmetric product term, (r_0_^2^r_1_^2^ r_2_^2^ r_3_^2^…), dominates Equation (4). Indeed, below their Equation 3.28, this is precisely the limit that Mathis et al. are considering. For r_i_ < 3 or so, their analysis is invalid in its prediction of a geometric series of periods. On the other hand, Figure 5 in their paper very clearly illustrates that for small “contraction factors” (i.e. large scale ratios) the Fisher information does a poor job in approximating the decoding error. So this means that the prediction of a geometric series of periods is based on a tenuous analysis with an uncertain range of validity. The optimal one dimensional grid in our work is perched near the edge of the estimated range where their analysis appears to valid. *Thus Mathis et al. predict a geometric scaling of the grid system only when the scale ratios r*_*i*_
*are large, while we know that these ratios are O(1) from experiment. What is more, their equations explicitly predict a hierarchy of scale ratios (Equation 6 above) in the O(1) regime.*

We then considered the possibility that the Fisher information approach of Mathis et al. could be rescued in two dimensions. Translating everything for two dimensional lattices would be a formidable work, so we contented ourselves with the following observations: (1) in two dimensions the Fisher information would scale the same way with the r_1_^2^ and so would take a similar form to Equation (4) above in terms of these variables, and (2) the constraint expression for the number of cells in terms of the r_i_ would still be symmetric between the r_i_. In two dimensions, experiments have shown a geometric series of periods with a period ratio of ∼1.5. This is too small for the last, symmetric term in the Fisher Infomation dominate. *Thus in two dimensions the analysis of Mathis et al. cannot predict a geometric series of grid periods in the regime of O(1) period ratios that applies to the data*.

Our paper uses very simple, general assumptions to make a number of specific quantitative predictions that Mathis et al. do not. Specifically, we: (1) predict a constant grid scaling ratio (in a regime where their alternative theory predicts a hierarchy of scales), (2) predict the grid scale factor (which they explicitly state they cannot do), (3) explain the 2d triangular grid geometry (which they do not even try to do), and (4) predict the ratio of grid period to grid field width under specific assumptions (which they parametrize in terms of tuning curve widths, contributing to their inability to predict the grid scale factor). We additionally predict the expected number of modules, and estimate the number of cells required in the mEC to implement our proposed grid scheme (see our Discussion).

The extension to two-dimensional grids is certainly non-trivial. There are many regular two-dimensional lattices, and our paper shows that the triangular lattice is favored. This is in no way implied by a one-dimensional analysis. We were able to study the two-dimensional lattices because we developed an analytical calculation (presented in the Appendix) that greatly simplified the numerical analyses.

Within the context of specific models of grid to place cell transformations we also show effects on spatial coding of selectively lesioning grid modules. (The latter analyses have significant caveats arising from our lack of knowledge of the precise relation between grid and place cells – this an important component of the comments of Referee 1 and our corresponding edits.)

All of these results go beyond the idea of a geometric progression of scales that Mathis et al. arrive at in one dimension through an extensive and sometimes inconclusive numerical analysis coupled with a study of the Fisher Information in the grid system, subject to the caveats described above.

*(b) It is true that, as stated in the in the conclusion, in the work of Mathis et al. the optimal ratio depends on “the number of neurons per module and peak firing rate”. But the prediction of the optimal ratio here also varies over a wide range, depending on the assumption on the decoding scheme (and probably the shape of the tuning curves, assumption on the correlation between neurons etc.)*.

For the reasons stated above, our analysis has a wider range of validity that Mathis et al. (please see our Discussion).

We do not agree that the optimal ratio here varies over a “wide range”. It is quite remarkable to us that an extremely simplistic winner-take-all decoder and an optimal probabilistic decoder give optimal ratios that are so closely clustered. The results do not depend in detail on the shapes of tuning curves etc. (please see our response to comment 1). Concerning the roles of correlations between grid cells, as Reviewer 1 points out, there is now work by the Herz group (38) and by the Roudi group (19) that suggests that there are only weak noise correlations between grid cells that are not aligned and of the same period (we now cite this work in the fourth paragraph of the Discussion).

Whether our prediction is “tight” or not may here be a case of beauty being in the eye of the beholder. The art of doing theory often involves making the right assumptions about the relevant and irrelevant factors. We made simple general assumptions that lead to remarkable (in our view) predictions for the architecture that agree with experiment. A legitimate way to do theoretical neuroscience is to make informed assumptions, build a theory with these assumptions, and then use the match between predictions of the theory and data as guide to whether the assumptions are reasonable. Certainly, methodologically, this seems like a very reasonable way to proceed, and is well within the venerable tradition of theoretical work in the older field of physics.

Reviewer #3:

*This excellent paper uses a very simple principle for demonstrating that coding of grid cells is better than coding of place cells, and generates some postdictions following this simple principle. The basic idea is that grid cells act as a kind of “Base-b” representation of space, and it is shown that the representation is optimal when the base chosen is base e (2.71828…). From that, various postdictions follow (which conform nicely with known experiments). Specifically, grid cell modules have a constant scale ratio, which should be √*e *in the simplest model, and closer to the real experimental value (1.4) in a probabilistic model of the cells coding. Furthermore, there should be a certain optimal ratio between the grid field width and the spacing between grid points*.

*The paper interacts nicely the papers of the group of Andreas Herz, which deal with similar issues using Fisher information. I have no major concerns, as I think the paper is well written, deals with an important subject, looks sound mathematically, and has a nice treatment of relation to experimental data*.

Thank you for these remarks. Please see below for changes we have made in response to the specific comments.

*The only issue I would like to be dealt with is to make the Discussion more clear as to the relation between this paper and the papers from the Herz group (including the relevant recent one from 2015). Specifically, they have a treatment of the issue of grid cell coding through Fisher information, and it could be of value to connect the work performed here to their line of thought, at least minimally by adding some discussion to the paper (elaborating on the existing paragraph)*.

We have now elaborated on the connection to the work from the Herz group. Specifically, we have added discussion of their use of the Fisher information and their different formulation of a resolution constraint. Please see the Discussion. Please also see our response to the editor’s remarks, and our response to Reviewer 2.

*Another small question I am curious about is whether the winner-take-all decoder could be seen as a limit-case of the probabilistic decoder. But if that is the case, I do not completely understand the “leap” from* e *to 2.4*.

The winner-take-all decoder is not quite a simple limit case of the probabilistic decoder as we have formulated it. One way to think of it is to imagine the WTA decoder as an approximation that replaces the smooth posterior with a flat function that drops to zero outside its support. The slight difference in the optimal ratio arises technically from the truncation of the tails in the Gaussian posterior, and the flattening of the posterior inside the region of support. Compared to the Gaussian, a boxcar likelihood has less precision (because it spreads out uniformly rather than being concentrated on the center), but it also implies less possibility of ambiguity (because it has zero tails). So the WTA decoder chooses a more aggressive (larger) scale ratio that improves precision, without being penalized by increased ambiguity.

We explain this point in the paragraph of the subsection “Probabilistic decoder” that starts “Why is the predicted scale factor based on the probabilistic decoder somewhat smaller than the prediction based on the winner-take-all analysis? […]”. We have slightly extended this paragraph.